# Loss of tumor suppressors promotes inflammatory tumor microenvironment and enhances LAG3+T cell mediated immune suppression

Sara Zahraeifard[1,6], Zhiguang Xiao[1,6], Jae Young So[1,6], Abdul Ahad[1], Selina Montoya[1], Woo Yong Park[1], Trinadharao Sornapudi[1], Tiffany Andohkow[1], Abigail Read[1], Noemi Kedei[2], Vishal Koparde[3,4], Howard Yang[1], Maxwell Lee[1], Nathan Wong[3,4], Maggie Cam[3], Kun Wang[5], Eytan Ruppin[5], Ji Luo[1], Christine Hollander[1] & Li Yang[1]✉

Low response rate, treatment relapse, and resistance remain key challenges for cancer treatment with immune checkpoint blockade (ICB). Here we report that loss of specific tumor suppressors (TS) induces an inflammatory response and promotes an immune suppressive tumor microenvironment. Importantly, low expression of these TSs is associated with a higher expression of immune checkpoint inhibitory mediators. Here we identify, by using in vivo CRISPR/Cas9 based loss-of-function screening, that NF1, TSC1, and TGF-β RII as TSs regulating immune composition. Loss of each of these three TSs leads to alterations in chromatin accessibility and enhances IL6-JAK3-STAT3/6 inflammatory pathways. This results in an immune suppressive landscape, characterized by increased numbers of LAG3+ CD8 and CD4 T cells. ICB targeting LAG3 and PD-L1 simultaneously inhibits metastatic progression in preclinical triple negative breast cancer (TNBC) mouse models of NF1-, TSC1- or TGF-β RII-deficient tumors. Our study thus reveals a role of TSs in regulating metastasis via non-cell-autonomous modulation of the immune compartment and provides proof-of-principle for ICB targeting LAG3 for patients with NF1-, TSC1- or TGF-β RII-inactivated cancers.

Immune checkpoint blockade (ICB) has recently emerged as a new treatment option for multiple cancer types. However, it is challenged by low response, treatment resistance/relapse, and a lack of biomarkers for patient stratification[1–3]. Treatment failure and subsequent tumor progression are orchestrated by dynamic and sophisticated tumor-host interactions involving alterations of both the tumor genome or epigenome and the immune microenvironment[2,4,5]. Immunogenomics analysis of more than 10,000 tumors across cancer types

[1]Laboratory of Cancer Biology and Genetics, Center for Cancer Research, National Cancer Institute, National Institutes of Health, Bethesda, MD 20892, USA. [2]Collaborative Protein Technology Resource, Center for Cancer Research, National Cancer Institute, National Institutes of Health, Bethesda, MD 20892, USA. [3]Collaborative Bioinformatics Resource, Center for Cancer Research, National Cancer Institute, National Institutes of Health, Bethesda, MD 20892, USA. [4]Advanced Biomedical Computational Sciences, Frederick National Laboratory for Cancer Research, Leidos Biomedical Research, Inc., Frederick, MD 21701, USA. [5]Cancer Data Science Laboratory, Center for Cancer Research, National Cancer Institute, National Institutes of Health, Bethesda, MD 20892, USA. [6]These authors contributed equally: Sara Zahraeifard, Zhiguang Xiao, Jae Young So. ✉e-mail: yangl3@mail.nih.gov

revealed a correlation of genomic aberrations with different tumor immune landscapes[6]. Recently, single-cell and spatially resolved transcriptomics analysis deconvoluted data from large breast cancer cohorts and stratified them into nine ecotypes with unique cellular compositions and clinical outcomes[7]. In addition, loss of tumor suppressors (TS) such as PTEN, p53, and p73 alters the immune-cell composition[8,9]. Notably, loss of PTEN, in combination with p53 or ZBTB7a, resulted in a distinct immune-cell composition[8]. Of great interest, the adaptive immune system was found to be a major driver for tumor suppressor inactivation in in vivo CRISPR screens comparing mouse tumor models with and without adaptive immune selective pressure[10]. Therefore, patient stratification based on integrated genotypic-immunophenotypic analyses is likely necessary for successful immunological therapies.

What connects tumor intrinsic genomic and epigenomic changes with the immune microenvironment is yet to be understood. We previously described a role for TGF-β signaling in tumor epithelium, where it has an anti-inflammatory role and functions as a regulator of the immune microenvironment[11–13]. Loss of p53 or p73, a member of the p53/p63/p73 family in breast cancer cells was reported to drive systemic inflammation and increased tumor-associated macrophages (TAM)[9,14].

TSs are conventionally known as tumor cell autonomous transcriptional and signaling regulators that negatively modulate cell cycle and survival to counteract the growth-promoting activity of oncogenes[15]. In addition to the mutation of TS genes, such as TP53, PTEN, and APC that predispose to cancers[16–18], the expression of TSs can be inhibited or silenced through epigenetic mechanisms (e.g., promoter hypermethylation)[19–21]. Recent data suggest that distinct genomic alterations cannot fully explain the emergence or expansion of therapy-resistant cancer cells. Rather, transcription reprogramming is a critical driver of tumor heterogeneity and evolution[22–27]. It remains to be investigated whether TS inactivation is a critical regulatory mechanism for the intricate relationship between the genomic features and the adaptive landscapes of tissue ecosystems.

Here we report that the inactivation of specific TSs such as NF1, TSC1, or TGF-β RII (TβRII) promotes a non-cell-autonomous inflammatory response through the IL6-JAK pathway and enhances LAG3+ T cell-mediated immune suppression. ICB targeting LAG3 and PD-L1 in triple-negative breast cancer (TNBC) mouse models with NF1, TSC1, or TβRII deficient tumors showed significant efficacy. We propose that the inactivation status of these three TSs should be one of the critical factors in ICB decision-making.

## Results

### The association of TSs with immune modulators and infiltrates in breast cancer patients

Emerging evidence suggests that TSs have a significant impact on the host immune landscape and responses[7–12,14]. We thus investigated the association of TS expression with immune modulators using the TCGA and the METABRIC datasets. We first curated 102 TSs from publications, which include the classic TSs, metastasis suppressors, and epigenetic suppressors (Table 1), and 45 immune modulators that include the negative immune checkpoint mediators (immune inhibitors), immune modulating cytokines and chemokines (Table 2). A matrix analysis of the TSs and immune modulators with all samples in the TCGA-BRCA dataset revealed a surprising negative correlation between the loss of TSs such as TSC1, BRCA1, NF1, APC, PTEN, and RB and key immune inhibitors including LGALS9, PD-1 (PDCD1), LAG3, ICOS, TIGIT, CTLA4, and IDO1 (Fig. 1A, purple boxes, TS-neg-Immune). Furthermore, these immune inhibitors showed a positive correlation with other TSs such as INPP5D, RASSF1, and CASP8 (Fig. 1A, green boxes, TS-pos-Immune). These results were further confirmed by scatter plot analysis of mRNA expression of TSs vs key immune inhibitors (Fig. 1B). Similar correlation patterns of TSs and immune

modulators were also observed in the METABRIC dataset (Fig. S1A). We further investigated the correlation between TS promoter methylation, a well-established mechanism of TS silencing, and immune gene expression. We used the AURORA dataset that has both gene expression and DNA methylation from the same breast cancer patient[28]. Patients with reduced TS gene expression that are preferentially mediated by promoter methylation had shown increased levels of immune inhibitors (from the TS-neg-Immune list) (Fig. S1B, Pearson correlation coefficient <−0.15, and Fig. S1C). These results suggest a significant correlation of decreased TS expression, with increased immune inhibitory molecules in a subset of TSs.

We next questioned whether decreased TSs result in any changes in immune cell composition. We used the TCGA-BRCA dataset that was computationally deconvolved by CODEFACS[29]. The patients were first divided according to their expression levels of TSs from the TS-neg-Immune list in cancer cells (Fig. 1C, left). The low-TS-patients showed increased immune inhibitors such as PD-1, LAG3, CTLA4, TIGIT in Treg cells, and PD-1 in CD8 T cells, CSF1R and IDO1 in macrophages, and IDO1 in neutrophils (Fig. 1C, right). Consistently, analysis of a clinical scRNA-seq dataset that contains 31 breast cancer patients[30] also demonstrated that patients with low expression of TSs from the TS-neg-Immune list had more PD1+ LAG3+ CD4 T cells and CTLA4+ Treg as well as PD1+ LAG3+ CD8 T cells (Fig. 1D and Fig. S1D). No major difference in CD4 and CD8 effector memory T cells was found (Fig. S1E). Analysis of a second clinical scRNA-seq dataset[7] also revealed increased LAG3+ CD8 T cells in patients with low expression of TSs from the TS-neg-Immune list (Fig. S1F).

Surprisingly, the expression of TSs showed distinct patterns among breast cancer subtypes, in which the basal subtype had an evident decrease in a subset of TSs (Group 1) and an increase in a different TS subset (Group 2) in both the TCGA and the METABRIC datasets (Fig. 1E and Fig. S1G). Interestingly, an inverse expression pattern was observed for the LumA subtype, while the Her2+ and Lum B subtypes had more intermediate patterns of expression (Fig. 1E and Fig. S1H). Notably, 12 of the top 20 Group 1 TSs (Fig. 1E, pink highlight) were TSs that are negatively correlated with immune inhibitors (Fig. 1A, pink boxes), with expression levels displaying the lowest in basal subtype when compared with other subtypes (Fig. 1F). These results suggest that decreased expression of certain TSs is associated with an immune suppressive microenvironment, which could be used for patient stratification in ICB.

### sgRNA screening revealed the metastasis-repressive function of NF1, TSC1, and TβRII

We choose to focus on TNBC, as it constitutes the majority of the basal subtype, and importantly, there is high mortality, lack of targeted therapies, and low response rate to ICB, especially for patients with metastatic disease[31,32]. Using the highly metastatic 4T1 TNBC mouse model, we performed the in vivo sgRNA loss-of-function screening of the 102 TSs used for the clinical data analysis and used metastatic nodule counts as a readout (Table 1 and Fig. 2A). A Cas9-expressing cell clone (4T1-C1) was selected to ensure the genetic and cellular homogeneity of cells receiving the sgRNA library (Fig. S2A). The library representation was ~300X and full coverage of all sgRNAs in a tight lognormal distribution was established (Fig. S2B). DNA sequencing was performed for the tumor tissues, draining lymph nodes and lungs collected at 2, 3, and 4 weeks after mammary fat pad (MFP) injection of tumor cells (Fig. 2A). Tumors with sgRNA-induced gene insertion/deletion (indel) mutations showed an increased metastatic nodule count and increased nodule size (Fig. 2B, C), with no difference in primary tumor size (Fig. S2C). As expected, the number of sgRNA species from the primary tumor tissues decreased over time due to clonal extinction (Fig. 2D), with less than 30% of sgRNA species persisting in the late-stage primary tumors (Figs. 2D and S2D).

**Table 1 | List of 102 tumor suppressor genes (TSGs):**

| Number | TSG | Group | Number | TSG | Group | Number | TSG | Group |
|---|---|---|---|---|---|---|---|---|
| 1 | GATA3 | GI | 61 | KISS1 | G0 | 65 | BRCA1 | GII |
| 2 | INPP4B | GI | 62 | AICDA | G0 | 66 | BRCA2 | GII |
| 3 | KAT6B | GI | 63 | DCC | G0 | 67 | MSH2 | GII |
| 4 | ARID2 | GI | 64 | KLF17 | G0 | 68 | BRD4 | GII |
| 5 | APC | GI | | | | 69 | TET1 | GII |
| 6 | NF1 | GI | | | | 70 | HDAC2 | GII |
| 7 | BMP4 | GI | | | | 71 | MAPK14 | GII |
| 8 | TSC1 | GI | | | | 72 | IDH2 | GII |
| 9 | NCOR1 | GI | | | | 73 | FBXW7 | GII |
| 10 | SIRT1 | GI | | | | 74 | KDM1A | GII |
| 11 | CREBBP | GI | | | | 75 | DUSP12 | GII |
| 12 | SRCAP | GI | | | | 76 | CDKN2A | GII |
| 13 | MAP2K4 | GI | | | | 77 | WT1 | GII |
| 14 | ARID1A | GI | | | | 78 | MBD2 | GII |
| 15 | PPP2CA | GI | | | | 79 | DOT1L | GII |
| 16 | NISCH | GI | | | | 80 | DAPK1 | GII |
| 17 | HDAC7 | GI | | | | 81 | CDKN2B | GII |
| 18 | TXNIP | GI | | | | 82 | EZH2 | GII |
| 19 | CDH13 | GI | | | | 83 | DNMT3B | GII |
| 20 | THBS1 | GI | | | | 84 | DNMT1 | GII |
| 21 | TIMP3 | GI | | | | 85 | DNMT3A | GII |
| 22 | DKK3 | GI | | | | 86 | CDH1 | GII |
| 23 | DLC1 | GI | | | | 87 | BRD7 | GII |
| 24 | AKAP12 | GI | | | | 88 | SMARCA4 | GII |
| 25 | TGFBR2 | GI | | | | 89 | CHD7 | GII |
| 26 | RB1 | GI | | | | 90 | PTPN14 | GII |
| 27 | PTEN | GI | | | | 91 | CD82 | GII |
| 28 | TET2 | GI | | | | 92 | BRMS1 | GII |
| 29 | KMT2C | GI | | | | 93 | PRMT1 | GII |
| 30 | KDM6A | GI | | | | 94 | SMARCB1 | GII |
| 31 | PBRM1 | GI | | | | 95 | MEN1 | GII |
| 32 | EP300 | GI | | | | 96 | NF2 | GII |
| 33 | HIC1 | GI | | | | 97 | DRG1 | GII |
| 34 | DUSP1 | GI | | | | 98 | NME1 | GII |
| 35 | KAT2B | GI | | | | 99 | EHMT2 | GII |
| 36 | STK11 | GI | | | | 100 | ASXL1 | GII |
| 37 | LIFR | GI | | | | 101 | SERPINB5 | GII |
| 38 | CDC73 | GI | | | | 102 | GSTP1 | GII |
| 39 | CHD4 | GI | | | | | | |
| 40 | VHL | GI | | | | | | |
| 41 | MED23 | GI | | | | | | |
| 42 | GPR68 | GI | | | | | | |
| 43 | MGMT | GI | | | | | | |
| 44 | SMARCA2 | GI | | | | | | |
| 45 | PEBP1 | GI | | | | | | |
| 46 | IDH1 | GI | | | | | | |
| 47 | HAT1 | GI | | | | | | |
| 48 | PRMT5 | GI | | | | | | |
| 49 | CASP8 | GI | | | | | | |
| 50 | ARHGDIB | GI | | | | | | |
| 51 | FHIT | GI | | | | | | |
| 52 | RASSF1 | GI | | | | | | |
| 53 | KAT5 | GI | | | | | | |
| 54 | MBD1 | GI | | | | | | |
| 55 | CHD5 | GI | | | | | | |

**Table 1 (continued) | List of 102 tumor suppressor genes (TSGs):**

| Number | TSG | Group | Number | TSG | Group | Number | TSG | Group |
|---|---|---|---|---|---|---|---|---|
| 56 | INPP5D | GI | | | | | | |
| 57 | GSN | GI | | | | | | |
| 58 | KISS1R | GI | | | | | | |
| 59 | HDAC5 | GI | | | | | | |
| 60 | KAT2A | GI | | | | | | |

low expressed (GI) and high expressed (GII) TSGs in the basal subtype and low expressed (G0) in all subtypes.

**Table 2 | List of Immune modulators for correlative studies**

| Co-stimulators | Inhibitors |
|---|---|
| CD160 | BTLA |
| CD226 | CCR4 |
| CD244 | CD47 |
| CD247 | CSF1R |
| CD27 | CTLA4 |
| CD28 | CXCR4 |
| CD3E | HAVCR2 |
| CD40 | KIR3DL1 |
| CD40LG | LAG3 |
| CD47 | NT5E |
| CD48 | PDCD1 |
| CD70 | TGFB1 |
| CD80 | TIGIT |
| CD86 | PVR |
| CEACAM1 | CD274 |
| ICOS | PDCD1LG2 |
| ICOSLG | CD276 |
| PTPRC | VTCN1 |
| TNFRSF14 | IDO1 |
| TNFRSF18 | SIRPA |
| TNFRSF4 | LGALS9 |
| TNFRSF9 | |
| TNFSF14 | |

In the metastatic lungs, over 10% of the sgRNA species were detected at 2 weeks and increased to over 20% at 3 and 4 weeks (Fig. 2D). Notably, sgRNAs targeting *Tgfbr2*, *Pten*, and *Tsc1* were highly enriched in the lungs as early as 2 weeks (Fig. 2E, left). Of notice, multiple sgRNAs targeting the same gene, some with 3–4 independent sgRNAs, were enriched. In addition, there was a significant enrichment of sgRNAs targeting *Pten*, *Tsc1*, *Map2k4*, and *Nf2* in 3-week lungs (Fig. 2E, middle), after the excision of visible metastatic nodules that was performed to recover sgRNA species in slow-growing or dormant nodules. Similarly in the 4-week lungs, the sgRNAs targeting *Tsc1* and *Tgfbr2* were highly enriched (Fig. 2E, right, Fig. S2E, F). Consistent with publications[33,34], sgRNA for the metastasis suppressor gene *Nme 1–2* was also enriched in the lungs collected at 2, 3, and 4 weeks (Fig. 2E). Together, these data highlight the importance of conventional tumor suppressor genes (e.g., *Tsc1* and *Tgfbr2*) in repressing tumor metastasis and the dynamic nature of clonal persistence and expansion.

When sequencing the individual metastatic nodules (Fig. S2G and H), we unexpectedly found that *Tgfbr2* sgRNA was highly enriched and constituted ~40% of all nodules 3 weeks after tumor injection (Fig. 2F). Interestingly, *Tsc1* and *Nf1* sgRNA enrichment was notably higher in 4-week nodules than the *Tgfbr2* sgRNA (Fig. 2F). *Map2k4, Hdac7, EP300* and *Rassf1* were also enriched among over 100 TSs screened (Fig. 2F).

Not surprisingly, the 21 genes out of the 25-candidate gene list from this screening consistently matched with the basal-low list from the human data analysis (Fig. 2F right panel, and Fig. 1E).

The top three TS genes identified by the sgRNA screening were *Tsc1*, *Nf1*, and *Tgfbr2*. The abundance of sgRNAs targeting these TSs increased in primary tumor tissues over time (Fig. 2G), suggesting a clonal persistence and expansion of tumor cells with TβRII, NF1, and TSC1 deficiency. In the lungs, *Tgfbr2* sgRNA was enriched at 2 weeks but decreased at 3 weeks, then maintained and stabilized at 4 weeks. *Nf1* sgRNA increased rapidly in the lungs over the time course, whereas the *Tsc1* sgRNA remained at a steady level (Fig. 2G). In contrast, *Pten* or *Nf2* sgRNA levels were high in the early stages in the draining lymph nodes and lungs but dropped rapidly over time (Fig. 2G). These data point to a repressive role of NF1, TSC1, and TβRII in metastatic colonization.

## Deletion of genes encoding NF1 or TSC1 or TβRII increased metastatic colonization

We next verified our findings from the Cas9-based sgRNA library screening by using individual sgRNAs targeting *Nf1*, *Tsc1* or *Tgfbr2* in the 4T1 orthotopic model of metastasis. The *Nf2* sgRNA was used as a control as it was not enriched in the individual metastatic nodules (Fig. 2F, G) and it dropped rapidly over time in the lungs after the excision of the metastatic nodules (Fig. 2G). As expected, knockout (KO) of *Nf1* or *Tsc1* or *Tgfbr2*, but not *Nf2*, increased the number of spontaneous lung metastatic nodules (Fig. 3A, B) and increased the primary tumor size (Fig. 3C). These findings were confirmed using shRNA knockdown of *Nf1*, *Tsc1*, or *Tgfbr2* and increased lung metastasis (Fig. 3D, E), excluding possible clonal effects associated with sgRNA-induced TS KO.

To understand how the host immune system contributes to the observed tumor phenotype, we next compared our findings from nude mice with those from the immune-competent Cas9 transgenic mice. The significant increase in the number of metastatic nodules upon *Nf1*, *Tsc1*, or *Tgfbr2* deletion as compared to 4T1-C1was recapitulated using the Cas9 transgenic mice (Fig. 3F, Fig. S3A, B). Surprisingly, there were more metastatic nodules in each of the groups from Cas9 transgenic mice than in nude mice, even though the primary tumors were the same size (Fig. 3F, Fig. S3A, B), suggesting the adaptive immune system present in Cas9 transgenic, not nude mice, is compromised and subverted to support metastatic colonization. Using the Cas9 transgenic mouse line, we further examined the direct effect of TS deficiency on metastatic colonization using the experimental metastasis model (tail vein injection) of TSAE1 tumor cells that carry a P53 homozygous hotspot point mutation G809A (R270H) (Fig. S3C) and Her2 overexpression (Fig. 3G, H, Fig. S3D), two well-recognized breast cancer etiology factors. NF1, TSC1, and TβRII deficiency in TSAE1 tumor cells significantly increased lung nodule counts (Fig. 3I). Altogether, our work demonstrates that NF1, TSC1, and TβRII TSs repress metastatic colonization.

## The expression of NF1, TSC1, and TβRII were decreased in metastasis nodules in orthotopic mouse models

To determine whether these TSs are downregulated in vivo during spontaneous metastatic progression, we used two TNBC 4T1 and

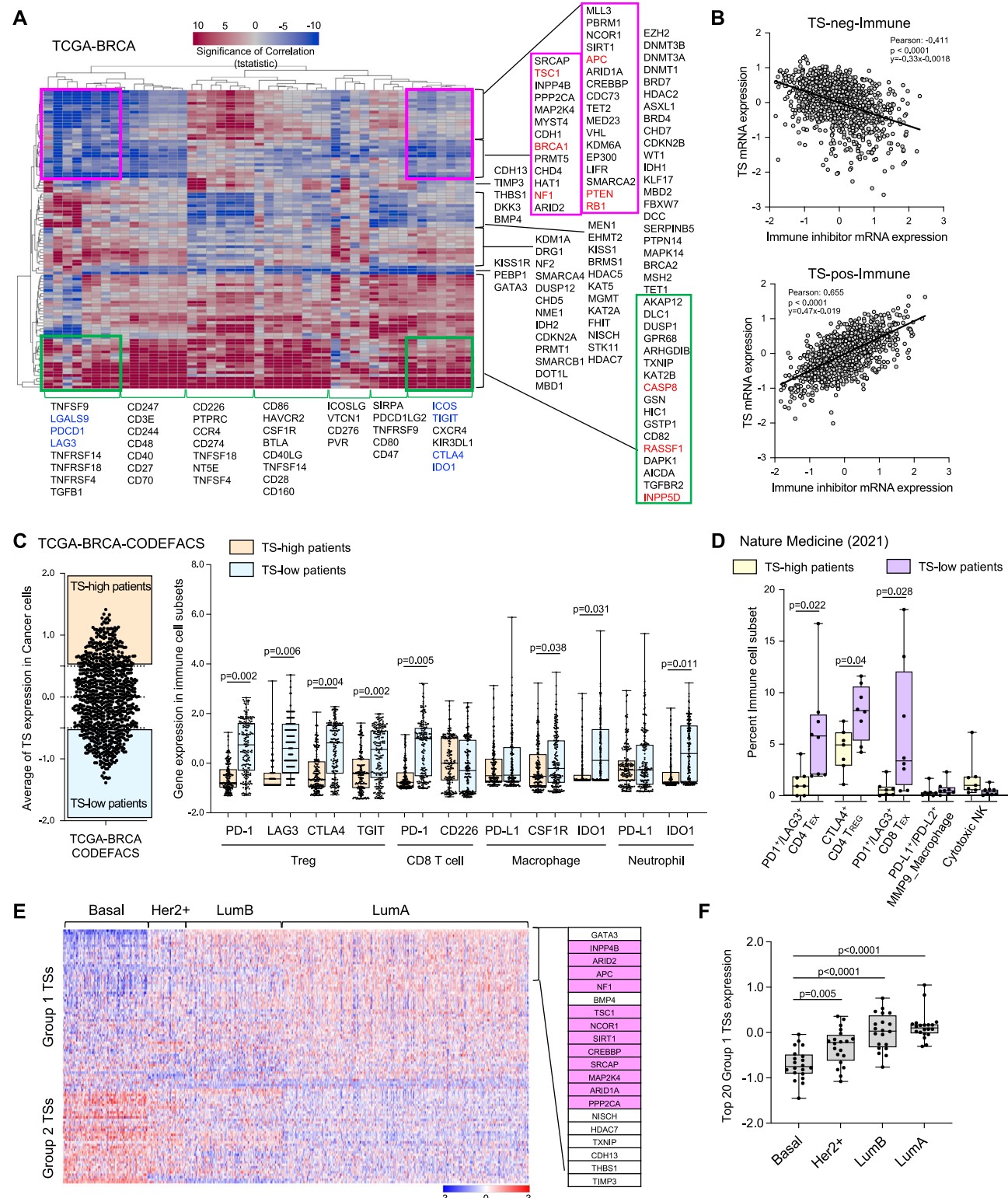

**Fig. 1 | The association of TSs with immune modulators and immune cell composition in breast cancer. A** Heat map of a matrix analysis showing the correlation of TS expression with immune modulators in the TCGA breast cancer dataset. blue: negative correlation; red: positive correlation; pink boxes: the TSs with negative correlation with immune modulators (TS-neg-immune); green boxes: the TSs with positive correlation with immune modulators (TS-pos-immune). Several oncogenes were used as a control. **B** Scatter plots of mRNA expression of TSs vs key Immune inhibitors ($n = 1082$). Upper panel: TS-neg-immune from the pink boxes; lower panel: TS-pos-immune from the green boxes. **C** Averaged expression levels of TSs from the TS-neg-Immune list for TCGA patients ($n = 867$). Left: deconvolved TCGA-BRCA dataset[29]; right: expression levels of immune modulators

(y-axis) in immune cell subsets between TS-high ($n = 197$) and TS-low patients ($n = 194$) (x-axis). **D** Percentages of immune cell subsets between TS-high ($n = 7$) and TS-low ($n = 8$) patients in single cell dataset[30]; $T_{EX}$ experienced T cells. **E** The expression of 102 TSs in four major subtypes of breast cancer patients. Right: list of top 20 Group 1 TSs; pink highlight: TSs overlap with the TS-neg-Immune list. **F** Expression of the top 20 Group 1 TSs in patients with different subtypes. All box plots show the 25th percentile, the median, the 75th percentile, and minimum/maximum whiskers. Statistical significance was determined by a two-tailed Pearson's correlation for (**B**); on-way ANOVA followed by Sidak's test for (**C**), (**D**), and (**F**).

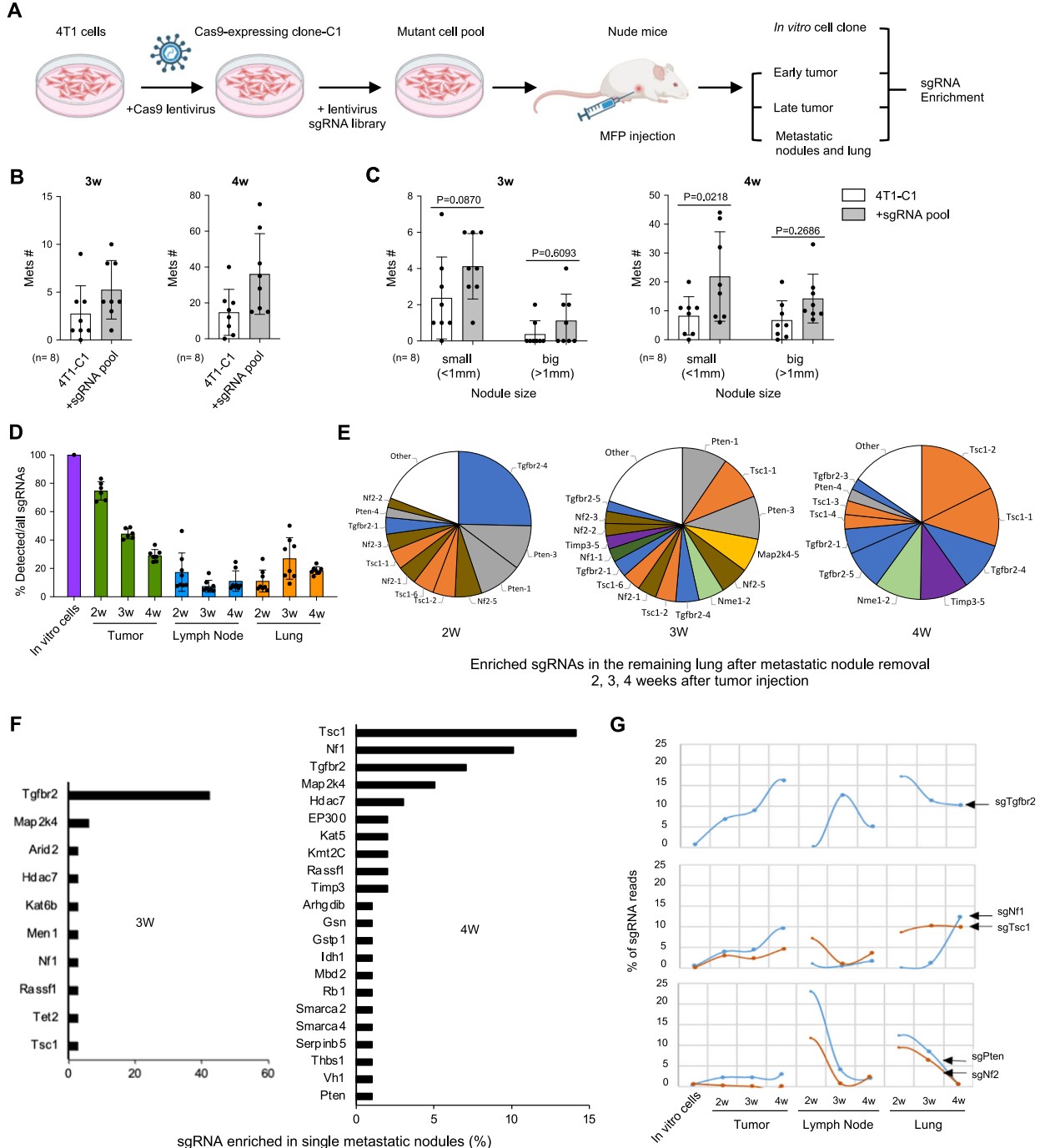

**Fig. 2 | sgRNA screening of candidate TS genes revealed NF1, TSC1, and TβRII with metastasis-suppressive function. A** Schematic loss-of-function screening for TS genes in 4T1 mammary tumor metastasis (created with BioRender.com). **B** Lung nodule counts at 3 weeks (left) or 4 weeks (right) after tumor cell injection and **C** Lung nodule counts with different sizes from immune-deficient mice received tumor cell injection. The number and size of tumor nodules on lung surfaces were counted under a dissecting microscope. *n* = 8 per group. **D** Representation of sgRNA library at different stages of tumor growth and metastasis. Number of sgRNA species in cells before transplantation, different stages of primary tumor, lymph node, and lung during tumor evolution. **E** Pie charts for the most abundant

sgRNAs in the lung after 2, 3, and 4-week sgRNA library-mediated cell injection. **F** Enriched sgRNAs in the lung nodules at 3 and 4 weeks after cell transplantation. Individual tumor nodules were taken out from the lungs, PCR amplified the sgRNA sequence, and examined through Sanger sequencing. **G** Dynamic evolution of sgRNAs indicated during tumor growth and metastasis. All sgRNAs with ≥2% of total reads are plotted individually. The remaining lung indicates most of the nodules were picked out before the evaluation. *n* = 8 per group. All bar graphs show mean ± s.d. Statistical significance was determined by two-tailed Student's *t*-test for (**C**).

EMT6 orthotopic models of breast cancer metastasis. We found that the protein expression of NF1, TSC1, and TβRII, but not NF2 and PTEN was downregulated in metastatic nodules compared to the primary tumor tissues (Fig. 4A, B). This observation was further verified by

immunofluorescence staining (Fig. 4C). These data demonstrate that the expression of NF1, TSC1, and TβRII was downregulated in metastatic progression, consistent with the human correlative analysis (Fig. 1E).

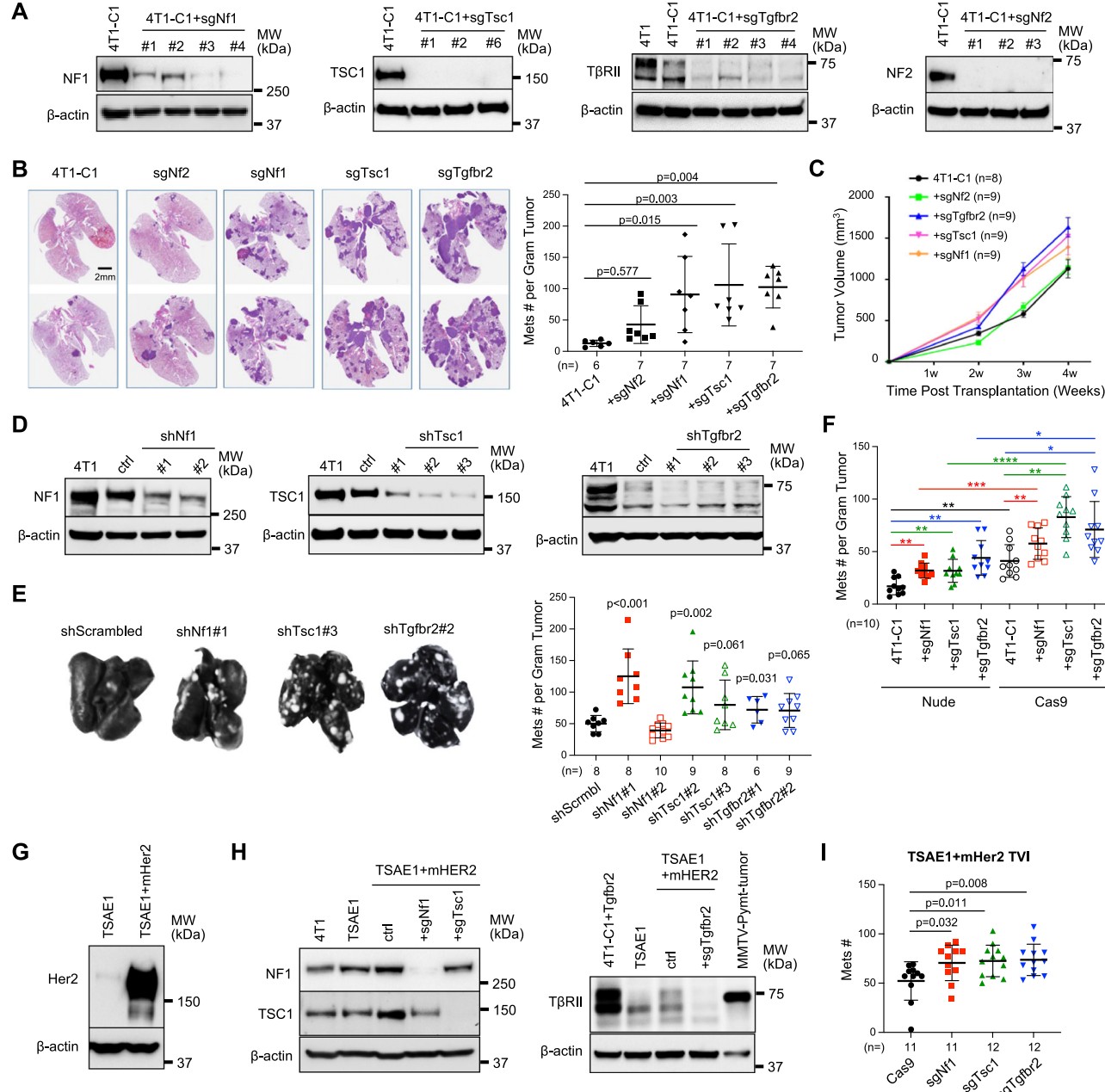

**Fig. 3 | Deletion of *Nf1*, *Tsc1*, or *Tgfbr2* increased metastatic colonization.**
**A** Western blot evaluation of individual sgRNAs induced *Nf1*, *Tsc1*, and *Tgfbr2* deletion, as well as *Nf2* KO. 3–4 sgRNAs were selected for each gene from the sgRNA library and used for validation. The Western blots were repeated for the cells with most significant KO for each TS gene and showed similar results. **B** Representative H&E staining of the lungs and number of metastatic nodules. Lung surface metastasis normalized to tumor mass of nude mice. *n* = 7 per group. **C** Primary tumor growth in nude mice transplanted with 4T1-C1 cells with sgRNA knockdown of *Nf1*, *Tsc1*, or *Tgfbr2* (*n* = 9 per group). **D, E** Western blot for shRNA knockdown of *Nf1*, *Tsc1*, or *Tgfbr2* (**D**), and representative Indian ink staining of the lungs (**E** left) and metastatic nodule counts (**E** right) (*n* = 10 mice each group). **F** Number of lung metastatic nodules (normalized to primary tumor weight) comparing nude and Cas9 transgenic mice that received MFP injection of tumor cells with sgRNA mediated *Nf1*, *Tsc1*, or *Tgfbr2* deletion. **G–I** TSAE1+mHer2 mouse model: Cas9 transgenic mice received TVI of TSAE1 tumor cells with sgRNA mediated *Nf1*, *Tsc1*, or *Tgfbr2* deletion: Western for Her2 overexpression in TSAE1 tumor cells (**G**); Western for sgRNA KO of *Nf1*, *Tsc1*, or *Tgfbr2* (**H**); metastatic nodule counts from lungs (*n* = 10 mice each group) **I**. Statistical significance was determined by one-way ANOVA followed by Sidak's test for (**B**); two-tailed Student's *t*-test for (**E**), (**F**), and (**I**). All graphs show mean ± s.d. *p < 0.05; **p < 0.01; ***p < 0.001. Exact *p*-values are provided in a source data file for (**F**).

## Deletion of *Nf1*, *Tsc1*, or *Tgfbr2* resulted in tumor cell-autonomous inflammatory reprogramming mediated by JAK-STAT3/6

We then performed RNA-seq and ATAC-seq to investigate whether there are shared downstream molecular alterations resulting from the three TS loss (Fig. 5A, Fig. S4A–E). RNA-seq analysis revealed 127 differentially expressed genes that were shared among NF1-, TSC1-, or TβRII-deficient 4T1 tumor cells compared to the 4T1-C1 control (Fig. 5B). The ATAC-seq data analysis identified 463 differential ATAC peaks common for all three TS KO vs control with consistency within duplicates and statistics (Figs. 5C and S4E). The integration of the RNA-seq and ATAC-seq data revealed that several key inflammatory genes were affected by chromatin accessibility upon TS loss. The promoter regions of JAK3 and IL6 genes showed clearly increased ATAC peaks

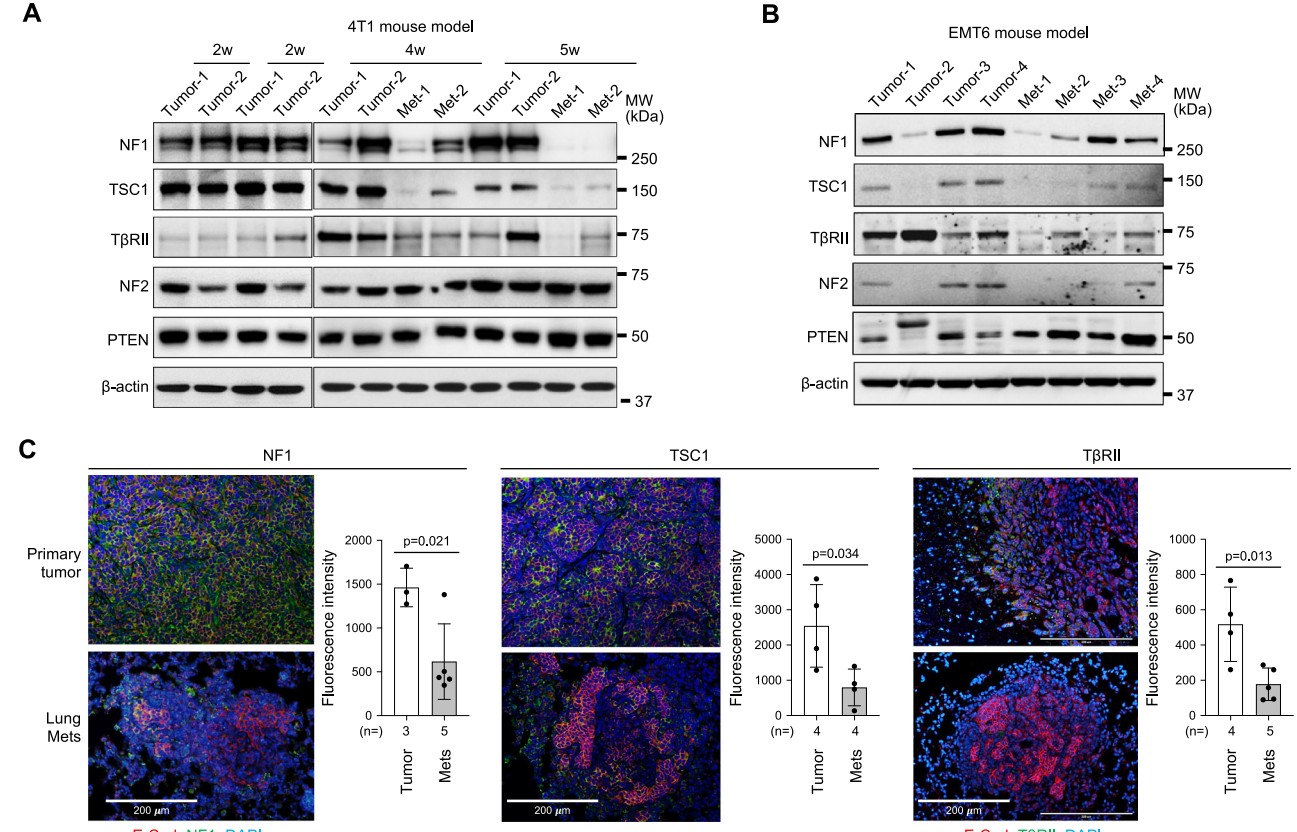

**Fig. 4 | NF1, TSC1, and TβRII were decreased in metastasis nodules in TNBC mouse models. A, B** Western blot of the protein extractions from mouse primary tumor and lung metastasis in 4T1 and EMT6 syngeneic orthotopic mouse models. The Western blots were repeated for both models with additional samples and showed similar results. **C** Immunofluorescence staining of NF1, TSC1, and TβRII in the primary tumor and lung metastasis collected at 5 weeks after 4T1 tumor cell injection. Representative images are shown. Quantitative data on fluorescence intensity are on the right. Statistical significance was determined by two-tailed Student's *t*-test for (**C**). All graphs show mean ± s.d.

for all three TS KOs (Fig. 5D). In contrast, the ATAC peaks were decreased at the CCL2 promoter in the three TS KOs (Fig. 5D). Integrated analysis of enrichment of transcription factor (TF) motifs (Motif match score >0.6) (Homer) vs frequency (Hocomoco-Fimo (1e-5) hits) identified TF candidates, among which STAT6 was one of the highest ranked (Fig. S4F). These data suggest loss of these TSs altered chromatin accessibility and transcriptional regulation of the IL6-JAK3 inflammatory pathway.

Consistent with the ATAC-seq and RNA-seq data, there was an increased expression of IL6, JAK3, and IDO1, another inflammatory mediator, and a decreased expression of CCL2 by q-PCR (Fig. 5E). The increased protein levels of IL6, JAK3, and IDO1 were further validated by ELISA and Western (Fig. 5F, left three panels) and decreased in CCL2 by Cytokine Array (Fig. 5F, right two panels). In addition, the phosphorylation of STAT3, STAT6, and JAK3 was increased in the TS KO cells compared to the controls (Fig. 5G). Notably, JAK3 not only exhibited an increased ATAC peak in the promoter region (Fig. 5D) and increased mRNA expression (Fig. 5E), but also increased phosphorylation (Fig. 5G), which could be activated by IL6 through autocrine signaling. Consistently and as shown in Fig. 5F, IL6, a key regulator of the JAK-STAT pathway, was significantly increased in all three TS KO tumor cell lines. To further dissect this signaling mechanism, we treated the cells with an IL6 neutralizing antibody, the pan JAK inhibitor Tofacitinib, and the JAK3 inhibitor Ritlecitinib. These treatments decreased the phosphorylation levels of STAT3 and STAT6 (Fig. S4G), as well as the production of IL6 and IDO1 (Fig. 5H) in these TS-deficient but not the control cells. Thus, loss of genes encoding NF1, TSC1, or TβRII resulted in tumor cell-autonomous inflammatory

reprogramming through the activation of the JAK-STAT3/6 pathway and resulted in increased production of IL6 and IDO1.

**Increased LAG3+ CD8 T and CD4 T cells resulting from NF1 or TSC1 or TβRII deficiency**

How distinct genetic alterations in tumors affect the composition of the immune landscape is currently unclear[8]. The changes in inflammatory and immune-regulatory pathways in TS-deficient tumor cells likely cause non-cell-autonomous changes in the adaptive landscapes of the tissue ecosystems. We further performed RNA-seq of the primary tumors from Cas9 transgenic mice with intact immune systems (Fig. S5A, B), which allowed us to capture changes in the inflammatory/immune microenvironment cues. Of the 150 differential genes shared by NF1-, TSC1-, or TβRII-deficient tumors (Fig. 6A, B), the immune regulation ranked among the top affected pathways (Fig. 6C). Of great interest, TS-deficient primary tumors showed an upregulation of immune inhibitors including LAG3, PD-L1, PD-L2, PD-1, TIM3, IDO1, and downregulation of several members of the TNF family (Fig. 6D). In addition, CD45 was also increased in all three KO tumors (Fig. 6D). Aside from the common immune modulators shared by the three TS-deficient tissues, there were also those specific for each of the TS loss (Fig. 6D). These data suggest that TS-deficient tumors elicit the formation of a more immune-suppressive tumor microenvironment.

We further investigated the immune cell phenotype, function, and spatial distribution in the TS-deficient tumors from the Cas9 transgenic mice using CYTEK (Fig. S5C, and Supplementary Table 1), a spectral flow cytometry, and CODEX (Supplementary Table 2), a

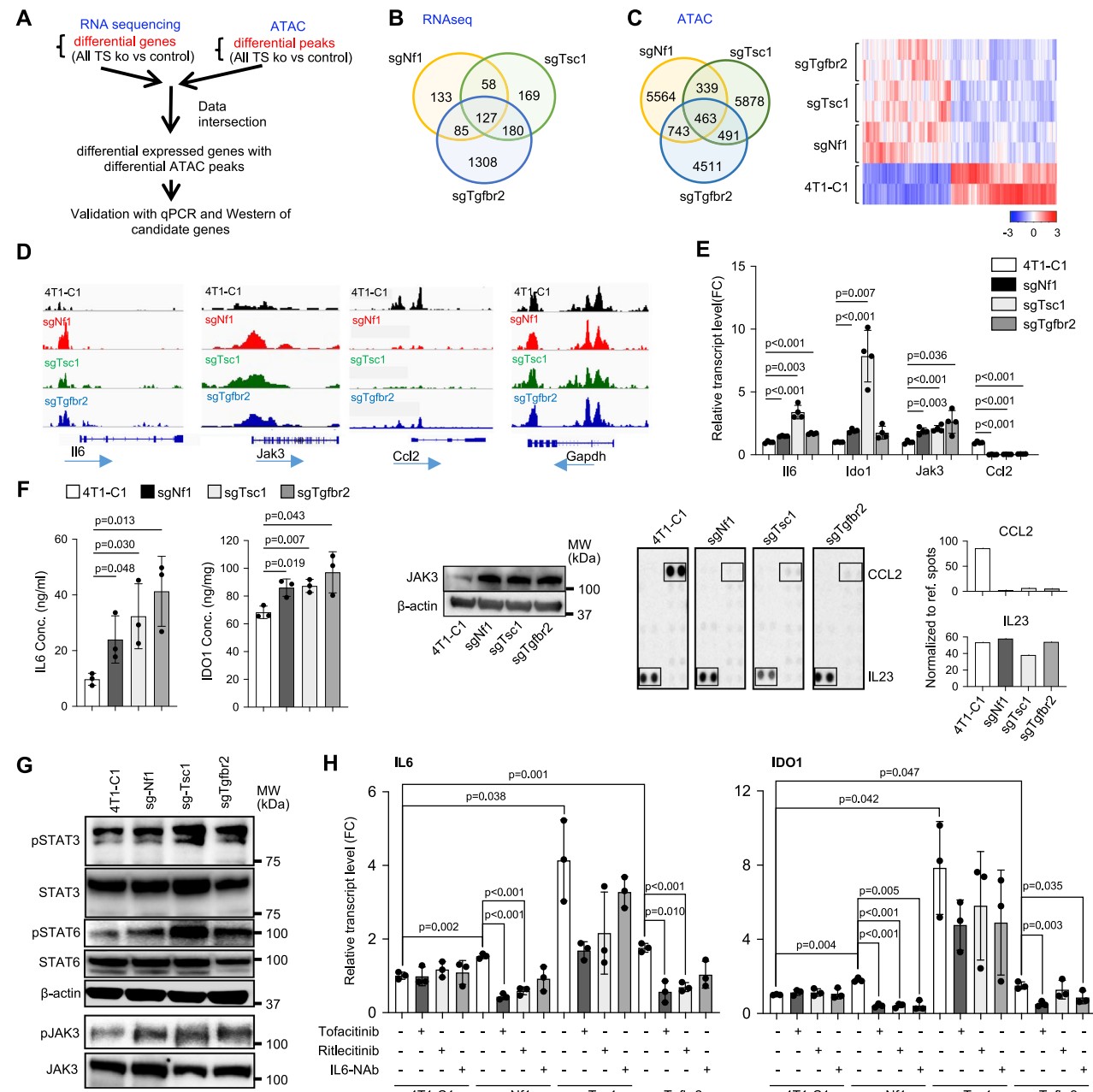

**Fig. 5 | Deletion of *Nf1, Tsc1*, or *Tgfbr2* resulted in tumor cell-autonomous inflammatory reprogramming by JAK-STAT3/6. A** schematic analysis for ATAC and RNA-seq datasets. **B** RNA-seq Venn diagram showing overlapped genes from in vitro cultured 4T1 cells with sgRNA mediated deletion of *Nf1, Tsc1, or Tgfbr2* compared with control 4T1-C1 cells. Cut-off: FC > 1.5, *P* < 0.05. **C** ATAC-seq Venn diagram (left) and heatmap (right) showing common differential peaks for all three TS KO vs the control cells. Cut-off: Differential ATAC peaks *p* < 0.01, FC < −2 and >2. **D** Increased ATAC peaks for IL6, JAK3 and decreased ATAC peaks for CCL2 that are common for all three TS KO cells, Gapdh as control. **E** Relative expression (RT-PCR) of IL6, JAK3, IDO1, and CCL2 comparing sgNf1, sgTsc1 or sgTgfbr2 vs control (*n* = 4).

**(F)** IL6 (*n* = 3) and IDO1 (*n* = 3) ELISA (left two panels), JAK3 Western, and CCL2 Cytokine Array and quantitative data (right two panels, IL-23 as control) from sgNf1, sgTsc1 or sgTgfbr2 compared with control. **G** Western blots of pSTAT3, pSTAT6, pJAK3 and JAK3 comparing sgNf1, sgTsc1 or sgTgfbr2 4T1 cells with C1 control. **H** RT-PCR of IL6 or IDO1 of cells treated with a pan JAK or a JAK3 specific inhibitor or IL6-neuAb (*n* = 3). All graphs show mean ± s.d. Average of each biological replicate was plotted, and statistical significance was determined by two-tailed Student's *t*-test for (**E**), (**F**), and (**H**). *\*p* < 0.05; *\*\*p* < 0.01. Exact *p*-values for (**E**), (**F**), and (**H**) are provided in a source data file.

multiplex fluorescence cytometric imaging. We observed an evident increase in LY6G+ neutrophils and a decrease in F4/80 macrophages (Fig. 6E). Furthermore, there was a decreased iNOS production in macrophages, increased PD-L1 expression in monocytes, and increased IDO1 expression in LY6G+ neutrophils and CD11b+ CD11c+ dendritic cells (cDCs) in TS-deficient tumors (Fig. S5D). We next sought to further dissect the cause-effect roles of LY6G+ neutrophils in the phenotype of NF1-, TSC1-, or TβRII-deficient tumors and focused on

the TSC1 KO model for functional studies since all three TS KO tumors showed a consistent increase in LY6G+ neutrophils. We depleted LY6G + neutrophils in both immune-competent and nude mice. Interestingly, LY6G+ neutrophil depletion decreased metastasis burden only in immune-competent mice but not in nude mice (Fig. 6F). These results suggest an important role for LY6G+ neutrophils and crosstalk with the adaptive immune system that influences the metastatic phenotype of TS-deficient tumors.

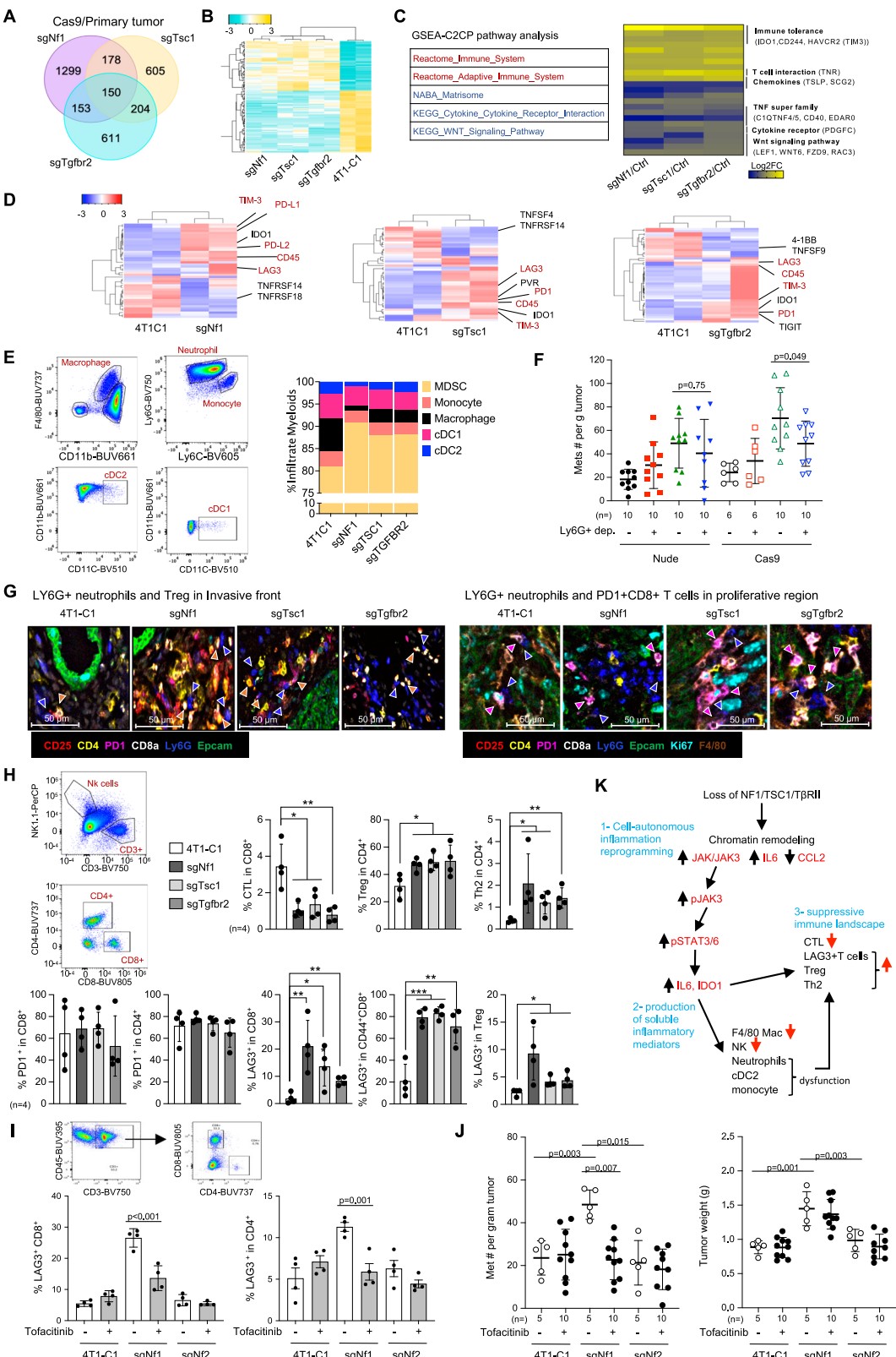

Of great interest, LY6G+ neutrophils were often found in the vicinity of PD-1+ CD8 T and Treg cells (Fig. 6G). The Treg cells were located mostly at the invasive front (Fig. 6G, Fig. S5E, F left panel), and PD-1+ CD8 T cells in the proliferative regions (Fig. 6G, Fig. S5E, F right panel). In analyzing the lymphoid cells, we noticed an increased number of CD4 T cells but no changes in CD8 T cells and fewer NK cells and B cells (Fig. S5G). Importantly, we observed an evident decrease in

CTLs and an increase in Tregs and Th2 cells (Fig. 6H, upper panels). Unexpectedly, there was no difference in the number of PD-1+ CD8 or PD-1+ CD4 T cells, but an increase in LAG3+ CD8 T cells and LAG3+ CD4 T cells for all three TS deficient tumors (Fig. 6H, lower panels). The increased LAG3+ CD8 T cells were especially evident in the CD44+ CD8+ T effector subset, 70-80% of which were LAG3+ (Fig. 6H, lower panels). In addition, there were also more LAG3+ Treg cells in TS-

**Fig. 6 | The non-cell-autonomous role of NF1, TSC1, or TβRII through inflammation and immune regulation. A** Venn diagram for differential gene expression comparing the primary tumors with sgNf1, sgTsc1, or sgTgfbr2 with controls in Cas9 transgenic mice. Cut-off: FC > 2, *P* < 0.01. **B** Heatmap of differentially regulated genes shared by all three primary tumors with sgNf1, sgTsc1, or sgTgfbr2 compared with control tumors. **C** IPA analysis of 150 genes that are shared among the NF1, TSC1, or TβRII deficient 4T1 tumor cells. **D** Heat map for major altered immune modulators in 4T1 tumor cells with NF1, TSC1, or TβRII deficiency. The gene names are listed on the right. Red color indicates genes shared among NF1, TSC1, or TβRII deficient cells. **E** Flow cytometry of various myeloid subsets from primary tumors with NF1, TSC1, and TβRII deficiency vs C1 control, showing % of MDSCs, monocytes, macrophages, cDC1, and cDC2. **F** Metastatic nodule counts (normalized to tumor weight) upon depletion of Treg or Ly6G cells comparing the Cas9 transgenic vs nude mice that bear TSC1 deficient tumors. Graph shows

mean ± s.d. **G** CODEX imaging of immune cells from primary tumors with NF1, TSC1, or TβRII deficiency. The blue, orange, and pink arrows show LY6G+ neutrophils, CD25+ Tregs, and PD1+ CD8+ T cells, respectively. **H** CYTEK analysis of primary tumors with NF1, TSC1, or TβRII deficiency. Upper: percentage of CTL, Treg, and Th2; Lower: percentage of PD-1+ in CD8 and CD4 T cells as well as percentage of LAG3+ in CD8, CD44CD8, and Treg cells. Graphs show mean ± s.d. **I** % of LAG3+ T cells in tumors (*n* = 4) treated with Tofacitinib. **J** Metastatic nodule counts (normalized to tumor weight) and tumor weight with Tofacitinib treatment. **K** Schematic hypothesis showing how NF1, TSC1, or TβRII deficiency leads to altered inflammatory and immune suppressive microenvironment. All graphs show mean ± s.d. Statistical significance was determined by Wald test for (**A**); two-tailed Student's *t*-test for (**F**) and (**H**); one-way ANOVA followed by Sidak's test for (**I**) and (**J**). *p < 0.05; **p < 0.01; ***p < 0.001. Exact *p*-values for (**H**) are provided in a source data file.

deficient tumors (Fig. 6H, lower panels). For the causal role of JAK/STAT signaling in the increased LAG3+ T cells and enhanced metastasis resulting from TS loss, the animals bearing 4T1-C1, 4T1-sgNf1, or 4T1-sgNf2 tumors were treated with Tofacitinib, an FDA-approved pan-JAK inhibitor preferentially targets JAK3 and JAK1[35,36]. Tofacitinib decreased the number of LAG3+ T cells and reduced metastasis in 4T1-sgNf1 tumors but had no effect on 4T1-sgNf2 tumors (Fig. 6I, J). Together, our data suggest that NF1, TSC1, or TβRII inactivation resulted in a tumor cell non-autonomous immune suppressive microenvironment that is potentially mediated by LAG3+ CD8 T and CD4 T cells (Fig. 6K).

## Immune therapy targeting LAG3 and PD-L1 decreased metastatic progression of tumors with NF1, TSC1, and TβRII deficiency

We further investigated the expression of NF1, TSC1, and TβRII in human breast cancer using the TCGA dataset. First, decreased NF1 and TSC1 were evident in the basal subtype compared with all other subtypes, and a decrease of TβRII in all subtypes compared with the normal tissue (Fig. 7A). Unexpectedly, the decreased NF1, TSC1, and TβRII expression did not clearly correlate with DMFS in the basal subtype of breast cancer patients (Fig. 7B). However, a TS-Imm signature, which included the three downregulated TSs and the corresponding increased immune inhibitors such as LAG3, PD-1, PD-L1, PD-L2, TIM3, as well as IDO1 and FOXP3 (for Treg), and GATA3 (for Th2), was associated with decreased OS and RFS for TNBC patients (*p* = 0.008) and decreased OS and RFS trend for Her2+ patients (*p* = 0.056) (Fig. 7C). In contrast, the TS-Imm signature was associated with increased OS and RFS for the Luminal subtypes (*p* = 0.00046) (Fig. 7C). Additional analysis using the GOBO dataset showed similar results (Fig. S6A). Thus, the expression of NF1, TSC1, and TβRII and a particular set of immune modulators could be critical for patient subtype-based prognosis.

Tumors with NF1, TSC1, or TβRII deficiency have an immune suppressive microenvironment in which LAG3+ T cells are abundant. Therefore, an immune therapeutic strategy that targets LAG3+ may be particularly effective in NF1, TSC1, or TβRII-deficient tumors. Indeed, LAG3 expression was negatively correlated with NF1, and TSC1 levels (Fig. 7D, left), which was particularly evident in TNBC, especially for TβRII (Fig. 7D, right). We next treated the Cas9 transgenic mice that bear NF1, TSC1, or TβRII-deficient tumors with anti-LAG3 alone or anti-LAG3/PD-L1 neutralization antibodies plus Paclitaxel (Fig. 7E). The anti-LAG3/PD-L1/Paclitaxel treatment showed the most significant metastasis reduction from tumors with NF1, TSC1, or TβRII inactivation but not from the 4T1-C1 control tumor (Fig. 7E and Fig. S6B). Additionally, paclitaxel treatment in combination with anti-LAG3 alone or with anti-LAG3/PD-L1 did not reduce metastasis from tumors with NF2 inactivation, suggesting a specific response of TS-deficient tumors to immunotherapies (Fig. S6C). Immune cell profiling indicated that the combination treatment did not change the number of myeloid subsets or the expression of functional markers including PD-L1, TNFa, Arg1, or

iNOS (Fig. S6D), but it increased CD4 and decreased Tregs (Fig. 7F, upper panels). Importantly, the combination treatment increased the percentage of CD8 T cells and decreased the expression of LAG3 and PD-1 in CD8 T cells (Fig. 7F, lower panels).

The NF1-, TSC1-, or TβRII-deficient tumors expressed high levels of IDO1 (Fig. 6C, D). Therefore, we also treated the mice with an IDO1 inhibitor, Epacadostat, in combination with the anti-LAG3 neutralization antibody and Paclitaxel (Fig. S7A). Consistently, the combination treatment decreased metastasis in mice bearing tumors with NF1, TSC1, or TβRII deficiency, which was not seen in the 4T1-C1 control (Fig. 7G, and Fig. S7A). In characterizing the immune microenvironment, we found that the treatment decreased the percent of LAG3+ CD8 and LAG3+ CD4 T cells as well as Tregs (Fig. 7G, and Fig. S7B). There were also increased IFNg+CD8 and IFNg+CD4 T cells (Fig. S7C). Similar to the anti-PD-L1 treatment, the number of myeloid subsets was not changed (Fig. S7D), but there was decreased PD-L1 expression in monocytes across tumors with NF1, TSC1, or TβRII deficiency (Fig. S7D). These data suggest that targeting LAG3 and myeloid inhibitors such as PD-L1 or IDO1 reduced metastatic progression of tumors with NF1, TSC1, or TβRII deficiency. Together, our studies propose a strategy for patient stratification based on NF1, TSC1, or TβRII expression and LAG3 immunotherapeutic intervention for patients whose tumors exhibit decreased expression or loss of NF1, TSC1, or TβRII.

## Discussion

We propose a direct and causal inflammatory or tissue damage response upon loss of TSs in tumor cells, which results in the formation of a unique tumor microenvironment that can be targeted. Loss of NF1, TSC1, or TβRII induced tumor cell-autonomous inflammatory reprogramming through alterations in chromatin accessibility and elevated IL6-JAK signaling. At the same time, loss of NF1, TSC1, or TβRII promoted the production of soluble inflammatory mediators such as IL6 and IDO1, which resulted in a non-cell-autonomous immune-suppressive tumor microenvironment that is characterized by increased LAG3+ CD8 and CD4 T cells (Fig. 6H). These insights into TS states and the associated tissue immune landscape provide both a mechanistic understanding and rationale for patient stratification and ICB targeting LAG3. These findings have yet to be verified in other cancers or different breast cancer subtypes, as TSs vary among cancer types, with each exhibiting distinct TS loss and oncogene mutations/overexpression. Further, in addition to shared downstream molecules resulting from the loss of NF1, TSC1, or TβRII, we also identified factors unique to specific TS loss, which will be of future research interest. Lastly, dissecting the molecular mediators underlying TS regulation of the chromatin landscape should offer more mechanistic insight.

The integration of cell-autonomous and non-cell-autonomous changes upon TS inactivation provides a more nuanced understanding of TS function, which is distinct from the conventional model in which TS loss primarily drives oncogenic activation of pathways such as AKT,

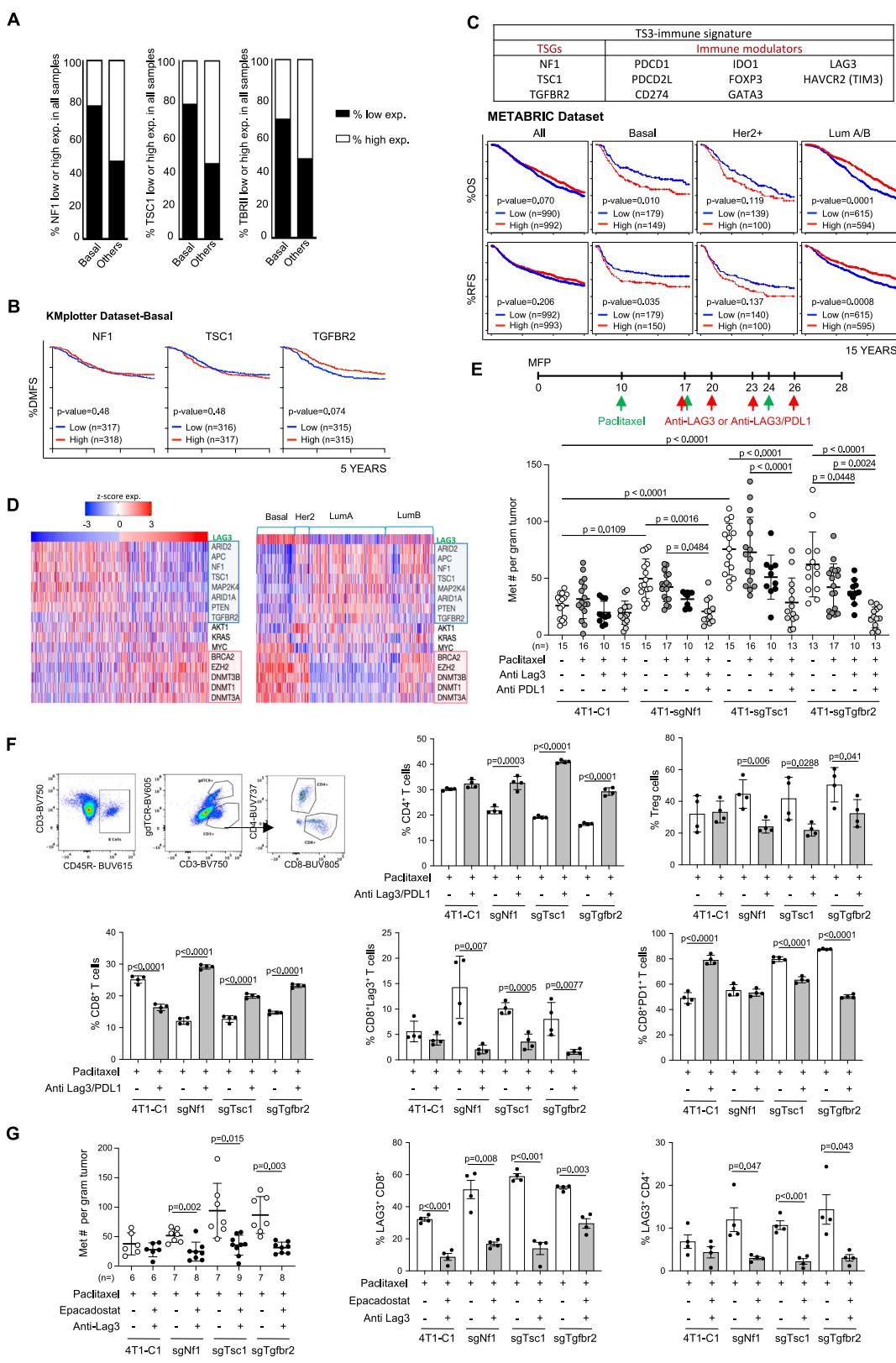

ERK, and mTOR, resulting in increased cell proliferation[15]. We have previously reported that loss or down-regulation of TGF-β signaling in epithelial cells induces an inflammatory and immune suppressive microenvironment through increased production of CXCL5 and recruitment of myeloid-derived suppressor cells[11,12]. Consistent with our findings, several TSs, including p53, APC, PTEN, and ARF, have been implicated in regulating inflammatory and immune

responses[8,37,38]. P53 and PTEN inactivation leads to an enhanced inflammation mediated by NF-kB[39–41], and releases factors that skew macrophage polarization to a more immune suppressive state in hepatic stellate cells[42]. Notably, non-cell-autonomous functions of oncogenes have been observed[43]. For example, MYC directly regulates CD47 and PD-L1 mRNA expression and suppresses innate and adaptive immune responses in T-cell acute lymphoblastic leukemia[44].

**Fig. 7 | Targeting LAG3 and PD-L1 in tumors with NF1, TSC1, or TβRII deficiency. A** Heatmap for the expression of NF1, TSC1, and TβRII in breast cancer subtypes (TCGA dataset). **B** Kaplan–Myer survival curve of breast cancer patients with the expression levels of NF1, TSC1, or TβRII in basal subtype breast cancer. **C** TS-Imm signature gene list, and the predicted Kaplan–Myer survival curve of all and basal, Her2+, Lum A/B subtype breast cancer patients. **D** Heatmap for the correlation of TS expression with LAG3 (left) and in subtypes of breast cancer (right), TCGA dataset. **E** upper: Schematic of combination treatment of anti-LAG3, anti-PD-L1, and Pac in 4T1 preclinical mouse model. lower: metastatic nodule counts normalized to tumor weight. **F** CyTEK analysis of primary tumors with NF1, TSC1, or TβRII deficiency from mice that were treated with anti-LAG3, anti-PD-L1, and Pac. Upper: percentage of CD4 and Treg cells; Lower: percentage of CD8, LAG3+ CD8, and PD-1+ CD8 T cells ($n = 4$). **G** Number of metastatic nodules per tumor weight in mice treated with anti-LAG3, IDO1 inhibitor, and Pac (left); LAG3+ CD8 and LAG3+ CD4 T cells from anti-LAG3 antibody, IDO1 inhibitor and Pac treatment ($n = 4$) (middle and right panels). Statistical significance was determined by logrank test for (**B**) and (**C**); one-way ANOVA followed by Sidak's test for **E**; two-tailed Student t-test for (**F**) and (**G**). All graphs show mean ± s.d.

Melanoma cell-intrinsic activation of β-catenin inhibits CCL4 expression and suppresses recruitment of dendritic cells and T cells, conferring immunotherapy resistance[45]. In breast cancer, JAG1-Notch signaling induces the expression of IL-1β and CCL2, facilitating the recruitment of tumor-associated macrophages in tumors[46]. These findings, along with our current study, highlight the importance of the non-cell-autonomous functions of TS loss and oncogene activation, as they have a direct impact on the efficacy of immunotherapies by promoting the formation of an immunosuppressive tumor microenvironment.

The downregulation or loss of NF1, TSC1, or TβRII or other TSs could be regulated transcriptionally through promoter hypermethylation, one of the most consistent epigenetic mechanisms in human cancers[19–21,47]. It is conceivable that epigenetic regulation provides the plasticity needed for adaptation to stress and microenvironment cues. In addition, epigenetic memory and inheritance offer advantages for tumor evolution during metastatic progression. However, the precise mechanisms are unclear and warrant further investigation. The epigenetic modulation of the tumor immune microenvironment has been reported to potentiate the efficacy of ICB[48]. However, epigenetic targeting strategies are often non-specific, which challenges their evaluation and necessitates a more systematic dissection of downstream transcription machinery. Perhaps the unique tumor microenvironment resulting from epigenetic-mediated TS loss offers an avenue for more specific targeting. Nevertheless, our studies support that the synergy between an altered immune microenvironment and the TS alterations that are directly acquired by cancer cells allows these cells to evade various tumor surveillance mechanisms and become metastatic.

We analyzed the link between the tumor cell-intrinsic properties of TSs and the associated immune microenvironment in multiple clinical datasets. A decreased expression of a subset of TSs correlated with increased expression of immune inhibitors and immune suppressive cells. Further, the in vivo loss-of-function Cas9-sgRNA screening identified NF1, TSC1, and TβRII as critical mediators in repressing inflammatory responses and breast cancer metastasis. Consistently, decreased NF1, TSC1, or TβRII correlated with a distinct immune signature and a worse metastasis-free survival. Importantly, ICB targeting LAG3 in combination with PD-L1 for tumors with inactivation of NF1, TSC1, or TβRII in preclinical mouse models demonstrated significant efficacy. TAMs are often correlated with poor prognosis and are recognized as important emerging targets for breast cancer immunotherapy[49,50]. However, this was not the case for TS inactivation in our studies, which suggests different targeting strategies are likely required for patients with TS loss or inactivation.

TS loss or inactivating mutations also play a role in tumor heterogeneity and therapy resistance[51,52]. For example, PTEN loss promotes resistance to T cell-mediated immunotherapy[53]. Tumor heterogeneity is another difficult challenge that limits therapeutic efficacy[54], which is likely mediated by dynamic and sophisticated tumor-host interactions[54]. The way forward lies in our ability to understand key genetic/epigenetic factors that could modulate the immune characterization of the tumor microenvironment. We propose that inactivation of TSs is critical in treatment design for patients whose cancers display inactivation or decreased expression of anti-inflammatory TSs.

## Methods

### Mice and cell lines

BALB/c and athymic nude mice (female, 6–8-week-old) were purchased from Charles River. FVB.129(B6)-Gt(ROSA)26Sor$^{tm1(CAG-cas9*,-EGFP)Fezh}$/J mice (stock No: 026481) were purchased from The Jackson Laboratory. These Rosa26-LSL-Cas9 knock-in mice were backcrossed to BALB/c mice for six generations to generate a congenic strain. Genotyping was performed on tail biopsies by PCR, with primer sequences CGGG CCATTTACCGTAAGTTAT, CCGAAAATCTGTGGGAAGTC, and AAGG-GAGCTGCAGTGGAGTA for mutant reverse, wild-type reverse, and common forward primers for the mutant and wild-type, respectively. 6–8-week-old CAS9+ female mice were utilized in the animal studies. All animal procedures reported in this study that were performed by NCI-CCR affiliated staff were approved by the NCI Animal Care and Use Committee (ACUC) and in accordance with federal regulatory requirements and standards. All components of the intramural NIH ACU program are accredited by AAALAC International. The maximal tumor size permitted by the protocol was 20 mm in any one dimension, and the maximal permitted tumor size was not exceeded. All mice were maintained at a 12-h light-dart cycle, 64–72 °F (18–22 °C) temperature, and 30–70% humidity condition and euthanized by following NCI CO$_2$ euthanasia protocol.

4T1 (Cat#, CRL-2539) and EMT6 (Cat#, CRL-2755) cell lines were obtained from the American Type Culture Collection, and TSAE1 cell line was kindly provided by Dr. Lalage M. Wakefield from the lab of cancer biology and genetics, National Cancer Institute. All cell lines were cultured in DMEM medium (Gibco) supplemented with 10% FBS (Gibco), 100 U/ml penicillin, and 50 μg/ml streptomycin (Sigma, St Louis). All cell lines were confirmed to be mycoplasma-negative by the MycoAlertTM Mycoplasma Detection Kit (Lonza).

### Human correlative studies

Publicly available datasets from patients with breast cancer (Kaplan–Meier Plotter, METABRIC, TCGA, and GOBO) were used to investigate the correlation of TS expression and Kaplan–Meier survival in breast cancer patients.

For the matrix analysis of the correlation between TSs and immune modulators, TCGA-BRCA samples were used in the regression analysis. For scattered plot analysis, the z-score normalized RSEM values were extracted from the TCGA-BRCA dataset. The average TS expression levels from the TS-neg-Immune list or from the TS-pos-immune were plotted with expression levels of immune inhibitors per patient. The correlation between TS promoter methylation and immune gene expression was analyzed using data from the AURORA US dataset (GSE209998 and GSE 212375)[28]. TS genes that showed a negative correlation between gene expression and promoter methylation ($r < -0.15$) were selected from the TS-neg-Immune list. Based on the average methylation levels of the selected TS genes, the patients were grouped and the expression of immune inhibitors was analyzed. For cell type-specific gene expression analysis, the computationally deconvolved TCGA-BRCA dataset by CODEFACS (https://zenodo.org/

record/5790343) was utilized[29]. The TS expression levels from the TS-neg-Immune list in cancer cells and the expression levels of immune modulators in immune cell subsets were determined for each patient.

For correlation between TSs and immune cell subsets, a scRNA-seq dataset from human breast cancer (https://lambrechtslab.sites.vib.be/en/single-cell) was used[30]. The cell annotations from the study, metadata of clustering all cells, and clustering per cell type were performed according to the published paper, and the cell count of each cell type was calculated per patient.

## Mouse models of tumor metastasis

For orthotopic metastasis, $2 \times 10^5$ 4T1 cells were injected into the mammary fat pad (MFP) #2 of the mice. Tumors were measured by caliper once a week and tumor volume ($V$) was calculated by the formula $V = (W2 \times L)/2$, where $W$ is the tumor width and $L$ is the tumor length. Primary tumors were weighed, and the number of lung metastases was evaluated after 4–5 weeks by Indian ink staining through the trachea or direct counting via a dissection microscope as stated in the legends. The number of lung metastases was normalized to tumor weight. For the second metastasis model, $5 \times 10^4$ TSAE1$^{Her2+}$ cells were injected through the tail vein (TVI). Statistical analysis was performed using GraphPad Prism Software (La Jolla, CA).

## sgRNA library design

Constitutive exons near the 5′ end of transcripts were identified using Illumina Human BodyMap 2.0 and NCBI CCDS datasets. sgRNAs were ranked by an off-target score using a metric that includes the number of off-targets in the genome and the type of mutations (distance from the protospacer-adjacent motif and clustering of mismatches) and those with the lowest off-target scores were selected. A custom sgRNA library was designed with 6 sgRNAs per gene to increase the chance of gene knockout. The library was screened at 270× representation. The sgRNA library includes 100 non-targeting (scrambled) sgRNA controls designed to have minimal homology to sequences in the mouse genome and 40 positive control sgRNAs targeting 40 genes where the loss of function of these genes is lethal to cells.

## Lentivirus production and transduction

The sgRNA library was synthesized using array synthesis and cloned as a pool into the lentiGuide-Puro vector (Addgene, #52963). The pre-designed shRNA plasmids were purchased from Millipore Sigma. The catalog numbers and sequences of shRNA were provided in Supplementary Table 3. 2.5 µg of lentiviral vector, 0.625 µg of PMD2.G envelope plasmid, and 1.875 µg of psPAX packaging plasmid were first added to 1 ml Opti-MEM, and 12.5 µl of the LTX reagent was then added to this mixture and incubated for 30 min at RT. The transfection complex was added to 80% confluent HEK293T cells in a 6-cm dish. The medium was replaced with fresh medium 24 h after transfection. Viral supernatant was harvested 48 and 72 h after transfection and stored at −80 °C.

## Cas9-expressing clonal 4T1-C1 cell line

The 4T1 cells were transduced with Lenticas9-blast lentiviral vector (Addgene, #52962). Successfully transduced cells were selected for with 4 µg/ml Blasticidin, then sorted as single cells into 96-well plates and cultured as clonal cell lines. Multiple clonal lines were established and Cas9 expression was evaluated by western blot. One clone, 4T1-C1, was used for relevant experiments.

## sgRNA library pool cell generation and cultivation

Pooled sgRNA library viruses were functionally titred and used to transduce 4T1 clonal cells at MOI = 0.3. The medium used contained 8 µg/ml polybrene. 24 h after transfection, the medium was replaced with fresh medium containing 3 µg/ml Puromycin. The medium containing the selected drugs was replaced every other day for a total of

7 days. The resultant drug-resistant cell lines were collected for sgRNA enrichment evaluation and in vivo transplantation.

## Individual sgRNA-induced gene knockout

sgRNA oligos were annealed and cloned into the lentiGuide-Puro vector (Addgene, #52963) using standard BsmBI protocols. Single sgRNA viruses were generated by using the same procedure as described for the library virus, and cancer cells were transduced at MOI = 10.

## sgRNA library readout by deep sequencing

Genomic DNA from cells and tissues was extracted using the DNeasy Blood & Tissue kit (QIAGEN, #69506). The sgRNA library readout was performed using two steps of PCR. PCR#1 included enough genomic DNA to preserve full library complexity using primers specific to the sgRNA-expression vector (lentiGuide-Puro), and PCR#2 used uniquely barcoded P7 and P5 adapter-containing primers to allow multiplexing of samples in a single HiSeq run. In the PCR#1 reaction, we used 3 µg of gDNA to capture 1000-fold representation of the library. The thermocycling parameters were: 95 °C for 5 min, 35 cycles of (95 °C for 35 s, 57 °C for 35 s, 72 °C for 35 s), and 72 °C for 10 min. All PCR was performed using Takara Hotstart Taq (ClonTech, #R007B). The PCR products were visualized with a 2% E-gel EX (Life Technologies) and cleaned up using the QIAquick PCR Purification Kit (QIAGEN, #28104) or QIAquick Gel Extraction Kit (QIAGEN, #28794). The second PCR products were pooled and then normalized for each biological sample before combining uniquely barcoded separate samples. The purified library was quantified with the Agilent 2100 Bioanalyzer (Agilent Technologies) and sequenced with MiSeq (Illumina). sgRNA sequence identification of individual nodules was evaluated using Sanger sequencing after PCR#1.

## RNA-seq

RNA was extracted from cells or tissues using the RNeasy Kit (QIAGEN). Tissue samples were flash-frozen and homogenized using a Precellys 24 tissue homogenizer (Bertin Instruments) for lysis before RNA extraction. RNA quality was evaluated using the Agilent 2100 Bioanalyzer (Agilent Technologies). RNA-seq samples were pooled and sequenced on HiSeq4000 using Illumina TruSeq mRNA Prep Kit (RS-122-2101) and paired-end sequencing.

The sequencing quality of the reads was assessed per sample using FastQC (version 0.11.5), Preseq (version 2.0.3), Picard tools (version 1.119), and RSeQC (version 2.6.4). Reads were then trimmed using Cutadapt (version 1.14) to remove sequencing adapters, prior to mapping to the mm10 mouse genome using STAR (version 2.5.2b) in two-pass mode. Overall expression levels were quantified using RSEM version 1.3.0 with GENCODE annotation M12. Limma (version 3.34.9) was used for differential expression analysis. For differential gene expression, $q \leq 0.05$ and absolute fold-change ≥1.5 were used to identify significantly altered genes.

## ATAC-seq

ATAC-seq was carried out according to Buenrostro et al.[55]. PCR was conducted for 10–11 cycles. Library purification was performed with magnetic double-sided bead purification and library size distribution was assessed using the Agilent 2100 Bioanalyzer (Agilent Technologies). Samples were pooled and sequenced on the NextSeq500 using V2 chemistry paired-end mode. The sequencing quality of the reads was assessed per sample and reads were then trimmed using Cutadapt (v 1.18) to remove sequencing adapters. Deduplication was performed using Picard's MarkDuplicate utility, prior to mapping to the mm10 mouse genome using Bowtie2 (v 2.2.6). Cyclic loess normalization was performed using csaw and followed by peak calling using Macs (v 2.1.2). The differential ATAC peaks with IDR < 0.05 were then identified using Limma. The transcription factor enrichment analysis was

performed using Homer (v 4.9.1) for common ATAC peaks shared by all three TS KO. For visualization, a BigWig file was generated using deeptools (v 3.1.2) and visualized on IGV.

## RT-qPCR

RNA was extracted (TRIzol Reagent Cat. 15596026, Invitrogen) and cDNA was synthesized (High-Capacity cDNA Reverse Transcription Kits, Applied Biosystems), with gene expression determined using SYBR Green-based qPCR. All primer sequences were provided in Supplementary Table 4.

## Western blotting and ELISA

Cells were plated at equal numbers following 24 h of starvation and 2–4 h of drug treatments (if mentioned). Supernatant was collected and total cell lysate protein was extracted using RIPA Lysis Buffer (Cat. R0278-50ML, Sigma) plus Complete protease inhibitor cocktail and phosphatase inhibitor (Roche). Tissues were flash-frozen with liquid nitrogen. Primary tumors and lungs were placed in vials (KT03961-1-002.2, Precellys) and grounded using a Precellys 24 tissue homogenizer (Bertin Instruments) for efficient tissue homogenization. Lung nodules were subjected to ReadyPrep™ Mini Grinders (BioRad). Protein concentration was measured using Pierce's BCA Protein Assay. An equal amount of total proteins was denatured, separated on NuPAGE 4–12% Bis-Tris protein gels, and transferred to a nitrocellulose membrane (Bio-Rad, #1704270). Membranes were incubated with primary antibodies against Cas9 (Diagenode, C15200203, 7A9), NF1 (abcam, ab17963), TSC1 (ThermoFisher, PA5-20131), TGF-βR2 (R&D System, AF532), NF2 (Cell Signaling, #6995, D1D8), Pten (Cell Signaling, #9559, 138G6), Caspase 3 (Cell Signaling, #9665, 8G10), pAKT (Cell Signaling, #9271), AKT (Cell Signaling, #9272), pERK (Cell Signaling, #9101), ERK (Santa Cruz, sc-514302, C-9), pmTOR (Santa Cruz, sc-293133, 59.Ser2448), HER2 (Cell Signaling, #2165, 29D8), CD40 (R&D System, AF440), pSTAT3 (Cell Signaling, #9145S, D3A7), STAT3 (Cell Signaling, #4904S, 79D7), pSTAT6 (Cell Signaling, #56554S, D8S9Y and 9361S), STAT6 (Cell Signaling, #9362S), pJAK3 (Cell Signaling, #5031S, D44E3), JAK3 (Cell Signaling, #8863S, D7B12), or β-actin (Santa Cruz, sc-69879, AC-15). Primary antibodies were diluted in 1:1000. Secondary HRP-conjugated anti-mouse (Cell Signaling, #7076S), anti-rabbit (Cell Signaling, #7074S) and anti-goat (Invitrogen, #31400) secondary antibodies were used with 1:2000 dilution and detection was done using an ECL reagent (Bio-Rad, #1705061). For protein concentration detection, supernatant and/or cell lysate protein extractions were examined using the Mouse IDO1(Indoleamine 2,3-dioxygenase 1) ELISA Kit (FineTest, EM0653), Quantikine™ ELISA Mouse IL6 Immunoassay (R&D Systems, M6000B), TGF-β1 Quantikine ELISA (R&D Systems, DB100B,), Proteome Profiler™ Array Mouse XL Cytokine Array Kit (Cat. ARY028, R&D Systems) according to manufacturer instructions.

## In vitro and in vivo mechanistic studies of the JAK-STAT pathway

After 24 h of starvation, cells were treated with Tofacitinib (10 nM, CP-690550-Selleckchem) or Ritlecitinib (500 nM, PF-06651600- Selleckchem) for pan-Jak or specific Jak3 inhibition, respectively. Also, InVivoMAb anti-mouse IL6 (MP5-20F3, 10 µg/ml) was used as an IL6 neutralizing antibody. Supernatant and cell lysate were collected between 30 min to 4 h for RT-qPCR or Western blot. For the in vivo study of JAK-STAT pathway, vehicle or Tofacitinib (5 mg/kg body weight) was intraperitoneally injected three times a week, starting from 5 days after the cancer cell injection.

## Immunofluorescence and immunohistochemical staining

Primary mammary tumors and the lungs were fixed in 10% neutral-buffered formalin overnight, embedded in paraffin, and sectioned at 6 µm. For immunofluorescence staining, sections were deparaffinized and rehydrated in ethanol, and antigen retrieval was performed in 0.01 mol/L sodium citrate (pH 6.0) for 20 min in a steamer. Sections

were incubated overnight at 4 °C with primary antibodies against E-Cadherin (1:50, BD Transduction Laboratories™, BD610181, C36), E-Cadherin (1:200, Cell Signaling, #3195, 24E10), NF1 (1:200, abcam, ab17963), TSC1 (1:200, ThermoFisher, PA5-20131), TGFβRII (1:100, R&D System, MAB532, C129502), followed by incubation with secondary antibodies goat anti-mouse Alexa Fluor 594 (1:500, A11005, Invitrogen), donkey anti-rabbit Alexa Fluor 488 (1:500, A21206, Invitrogen), donkey anti-rabbit Alexa Fluor 594 (1:500, A21207, Invitrogen), or donkey anti-rat Alexa Fluor 488 (1:500, A21208, Invitrogen) for 1 h at RT. Tissues were then counterstained with DAPI to visualize the nuclei. The samples were mounted in Vectashield (Vector Laboratories) and subjected to microscopy analysis (Olympus IX-81). For quantification, multiple regions of fixed size were selected, and fluorescence intensities of given tumor suppressors were quantified by ImageJ (https://imagej.nih.gov/ij/).

## Analysis of tumor immune infiltrate by CyTEK Aurora

A multicolor antibody panel covering myeloid to lymphoid cells was used in CyTEK analysis (Supplementary Table 1). Primary tumor tissues were harvested from Cas9 transgenic mice that received $2 \times 10^5$ 4T1-C1 and three TS KO cell injections after 28 days. Briefly, tumors were weighed, mechanically disrupted, and incubated in RPMI medium containing 0.015 g/mL of DNase I (Sigma, D5025-750KU), 0.012 g/mL Dispase (neutral protease, Worthington, LS02104), and 1 mg/mL of collagenase (Worthington, LS004154) for 40–45 min at 37 °C. Red blood cells were lysed using ACK buffer and 3–5 million cells resuspended in 100 µl PBS and stained with 1:1000 LIVE/DEAD™ Fixable Blue Dead Cell Stain Kit, for UV excitation (Thermo Fisher, L34962) for 30 min at 4 °C. Fc receptors were blocked for 5 min with anti-mouse CD16/32 antibody (BioLegend, #101320). Next, cells were surface stained with the corresponding antibody in 1:100 dilution (Supplementaru Table 1) cocktail prepared in FACS buffer. Following fixation, cells were permeabilized and intracellularly stained in 1:60 dilution (Supplementary Table 1) and samples were acquired in CyTEK Aurora software, and data was analyzed with FlowJo software (v10.6.0). Frequencies from total leukocytes (live CD45+ cells) were determined and absolute numbers per mg of tumor were calculated.

## CODEX immunostaining

Co-detection by indexing (CODEX), a multiplex immune fluorescence cytometric imaging technology for the simultaneous measurement of ~20–50 targets in a single tissue section, was carried out for immune cell profiling and spatial localization in the primary tumors comparing NF1-, TSC1-, and TGFβRII-deficient tumors with 4T1-C1 controls. CODEX 27 antibodies, reagents (including those for conjugation of custom antibodies), and instrumentation were purchased from Akoya Biosciences (Marlborough, MA). The information and dilution factor for each antibody were provided in Supplementary Table 2. OCT fresh-frozen tissue samples were sectioned onto poly-L-Lysine-treated coverslips. Fixation, staining, and CODEX assays were performed according to the manufacturer's recommendations. The signals were visualized by a Keyence BZ-X810 microscope with a CFI Plan Apo 20x/0.75 NA objective (Nikon). The acquired raw tiff images were transferred to the processing computer and processed using CODEX processor version 1.7.0.6 without segmentation. Nuclear staining from cycle 2 was used as a reference for focusing. Processed images were loaded into HALO 2D digital pathology analysis software (IndicaLabs) for analysis.

## In vivo myeloid cell depletion

For in vivo myeloid cell depletion, InVivoMab anti-mouse Ly6G (1A8; 25 µg/mouse) or IgG2a isotype control (2A3; 25 µg/mouse) antibodies were intraperitoneally injected every 2 days starting from day −1 of cell injection until mice were euthanized at day 28.

## Mouse models for combinations of chemotherapy and immune checkpoint blockade

For the anti-LAG3 and anti-PD-L1 combined immunotherapy, the 4T1 tumors were allowed to grow for 10 days until palpable, then mice were treated with Paclitaxel (6 mg/kg body weight, i.v. once a week) followed by i.p. injection of anti-PD-L1 (200 μg, clone 10 F.9G2; Bio-XCell) and anti-LAG3 antibody (250 μg, Clone C9B7W, BioXcell) every 3 days until mice were sacrificed at the end of the experiments. For the anti-LAG3/IDO1-inhibitor combined immunotherapy, the mice were administered the anti-LAG3 antibody (250 μg, Clone C9B7W, BioXcell) by i.p. injection every 3 days and the IDO1 inhibitor epacadostat (100 mg/kg body weight, INCB024360, Selleckchem,) by oral gavage once every other day for a total 5 doses until mice were sacrificed at the end of the experiments. Tumor phenotype and immune profiling were obtained as indicated above.

## Statistics & reproducibility

Data are presented as mean ± Standard Error (SE). Comparison between two groups was performed using a two-tailed $t$-test, unless otherwise indicated. Comparison between more than two groups was performed using one-way ANOVA followed by multiple comparisons with Sidac correction. Statistical analyses were performed using GraphPad Prism 7.0 software, and significance was defined as $p < 0.05$. No statistical method was used to predetermine the sample size. No data were excluded from the analyses, except for Fig. 5F, left panel, in which one replication was excluded due to a noted technical problem. All mice in vivo studies were randomized before cancer cell injection and drug treatment. Immune profilings using CyTEK were performed blindly by separate researchers. The phenotype of animals experiments was not evaluated blindly, and major results were verified by separate researchers.

## Reporting summary

Further information on research design is available in the Nature Portfolio Reporting Summary linked to this article.

## Data availability

The data supporting the findings of this study are available within the article and the supplementary information files are available from the corresponding author upon request. The raw and processed sequencing data from ATAC-seq and RNA-seq have been deposited in the GEO database and are publicly available under accession numbers GSE249967, (GSE249969) and GSE249971. All relevant source data for each figure are provided. Source data are provided with this paper.

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

## Acknowledgements
We thank Drs. Eva Perez Guijarro, Chi-Ping Day, and Brandi Carofino for critical reading of the manuscript. We thank Drs. William Telford and Gregoire Altan-Bonnet for their support in characterizing the immune cells. We appreciate Drs. Eytan Ruppin and Kun Wang for their support in human correlative studies. We are grateful for technical assistance from the sequencing core and animal facility. This work was supported by NCI intramural funding to Dr. Li Yang.

## Author contributions
S. Zahraeifard and Z.G. Xiao designed, planned, and performed most of the experiments, analyzed data, and wrote a draft of the paper. A. Ahad performed CYTEK and analyzed the data. J.Y. So, T. Andohkow, S. Montoya, W.Y. Park, and T. Sornapudi were involved in mouse models and protein extraction. V. Koparde, H. Yang, M. Lee, N. Wong, M. Cam, K. Wang, and E. Ruppin contributed to bioinformatics and statistical analysis. A. Read and J. Luo contributed to the sgRNA library design. N. Kedei performed the CODEX experiments. All Yang lab members participated in the discussion of experimental designs, data analysis, and editing of the manuscript. L. Yang initiated, organized, and designed the study, and supervised the overall project and the manuscript preparation.

## Funding

## Competing interests
The authors declare no competing interests.
