## [Peer Review File · Nature Communications]

Loss of tumor suppressors promotes inflammatory tumor microenvironment and enhances LAG3+T cell mediated immune suppressionREVIEWER COMMENTS

Reviewer #1 (Remarks to the Author):

The authors of this study investigated the role of tumor suppressor inactivation in the modulation of the tumor microenvironment and its impact on immune suppression mediated by LAG3+ T cells. By performing in vivo loss-of-function screening of 102 tumor suppressors in the highly metastatic 4T1 triple-negative breast cancer mouse model, the authors uncovered a novel mechanism in which inactivation of tumor suppressors promotes a non-cell-autonomous inflammatory tumor microenvironment.

This inflammatory microenvironment was found to enhance LAG3+ T cell-mediated immune suppression, providing valuable mechanistic insight into tumor progression and immune evasion. Additionally, the study offers a rationale for patient stratification and potential therapeutic strategies targeting LAG3 in immune checkpoint blockade therapy.

The concept of an inflammatory tumor microenvironment enhancing immune suppression, including LAG3+ T cell-mediated immune suppression, is not entirely novel. Previous studies have reported the involvement of an inflammatory tumor microenvironment in immune suppression and tumor progression. Furthermore, a growing body of literature demonstrating that tumor progression resulting from the downregulation of tumor suppressor genes can be immune-mediated, in addition to being driven by oncogene expression and increased proliferative activity. However, this work would make an excellent addition to this literature and the novelty of the study lies in the specific context of the inactivation of tumor suppressors (i.e. TNBC), the specific findings connecting inactivation of tumor suppressor genes to LAG3+ T cell-mediated immune suppression represent a novel contribution to the field.

The finding that immune therapy targeting LAG3 and PD-L1 decreased metastatic progression of tumors with NF1, TSC1, and TBR1 deficiency is also notable. By showing that targeting both LAG3 and PD-L1 can reduce metastatic progression in this specific context, the study provides valuable information for tailoring immune therapies and potentially improving patient outcomes.

The overall design and approach employed is principled and the methods, including for the generation and analysis of omics profiling data are suitable. Overall, this is excellent work and is well presented (except, surprisingly, for obvious typos – see below). Also, my comments are relatively minor.

Main Comments:

- It would be appropriate and useful to bring more context in the discussion (see my introductory comments). Prior studies have demonstrated associations between the overexpression of oncogenes and the development of an immunosuppressive microenvironment and should be cited/discussed. e.g. Casey et al. [1], where overexpression of MYC caused by the repression of TS genes was found to contribute to the development of an immunosuppressive TME by upregulating the expression of ICs CD47 and PD-L1. Or Spranger et al [2], where activation of the β -catenin signaling pathway, which can be triggered by the inactivation of tumor suppressor genes like APC, can lead to an immunosuppressive TME in melanoma. [1] (Casey, S. C., Tong, L., Li, Y., Do, R., Walz, S., Fitzgerald, K. N., Gouw, A. M., Baylot, V., Gütgemann, I., Eilers, M., & Felsher, D. W. (2016). MYC regulates the antitumor immune response through CD47 and PD-L1. *Science*, 352(6282), 227-231) [2] Spranger, S., Bao, R., & Gajewski, T. F. (2015). Melanoma-intrinsic β -catenin signaling prevents anti-tumour immunity. *Nature*, 523(7559), 231-235.
- The authors do not discuss limitations of their study. This oversight should be rectified.
- RNA-seq and ATAC-seq data have not been deposited in public repositories. It should be the case for transparency-sake and to align with open data standards. Failing to do so would go against the FAIR principles for data management and sharing and significantly reduce the long-term impact and interest in the study.

- While the authors have taken care to minimize off-target effects during library design (ranking sgRNAs by an off-target score using a metric), they do not mention including non-targeting (scrambled) sgRNA controls to account for potential off-target effects.

Minor comments

- It is not clear why the color scheme for TS-high vs TS-low patients changes between panels C and D on figure 1 (or E and F in Figure S1).
- Font size on the volcano plot Figure 5S is impossibly small
- There are some typos / incomplete sentences here or there. Check for instance:
 - o Line 164: sentence construction, missing word?
 - o Line 165 and other places: neutrophils
 - o Line 181: sentence construction
 - o Fig S5 legend: "balk" is presumably meant as "bulk"
- Just a comment, but it makes it much easier to review a manuscript when the legends are embedded with the figures (as you have done for supplementary figures).

Reviewer #2 (Remarks to the Author):

In the manuscript, "Loss of tumor suppressors promotes a non-cell-autonomous inflammatory tumor microenvironment and enhances LAG3+T cell mediated immune suppression" Zahraeifard and colleagues aim to understand how loss of expression of tumor suppressors contributes to an altered immune microenvironment and influences response to anti-cancer immunotherapy.

They use a combination of human datasets with cell line xenograft and syngeneic mouse tumor models to perform a CRISPR screen and identify loss of which tumor suppressors is associated with increased number of lung metastases. They then use RNA seq and ATAC seq to show that the genes altered by TS loss are in the STAT pathway. Using flow cytometry, CODEX and CYTEK on transgenic mice they find that TS deficient tumors are associated with increased LY6G+ neutrophils and LAG3+ T-cells. They then combine paclitaxel, anti Lag3 and PDL1 targeting therapy to show that response to this combination is increased in TS lost tumors compared to control.

The authors have tackled an important question: how does the loss of tumor suppressors affect the anti-tumor response. As this is a complicated multi-step process they have shown evidence from each layer of the 'system' to support their argument. However, their claim that "a direct and causal" [line 395] inflammatory or tissue damage response in response to TS loss is not supported by the data. Instead, what they show is several associations without clear mechanistic connection between each phenotype e.g. TSC1 loss and increased tumor growth and associated STAT pathway activation and associated LAG3+ T-cell infiltrate. They have not shown how TSC1 loss is necessary and sufficient to alter the chromatin landscape within tumor cells resulting in JAK-STAT pathway activation and consequent alteration in the tumor microenvironment. In fact, given that loss of TSC1, TGFBR2 and NF1 lead to common ATAC changes in IL6 and JAK3, is it instead more likely that a separate downstream activator is causally responsible for this change in expression program? One way to demonstrate that this is a specific effect causally related to tumor suppressor loss is to show that phenotype can be rescued. For example, for tumors with TSC1 loss, does mTOR inhibition blunt tumor growth by preventing differential expression of JAK-STAT pathway genes? Can they further show this is a specific effect to the loss of their selected tumor suppressors (e.g. NF2) by using their elegant negative control sgNF1? For example, if sgNF1 tumors are compared to sgNF2, does the JAK-STAT pathway still appear to be responsible for the difference in tumor metastasis burden? Is there also a difference in LAG3+ T-cell frequency? Do sgNF1 tumors respond less well than sgNF2 tumors to treatment with paclitaxel and PDL1/LAG3? This is an important question to determine the specificity of the TS loss phenotype.

The manuscript would be further improved by making the following minor revisions:

The authors have curated a list of 102 proteins and defined them as tumor suppressors. Each has a different function and many work in concert to inhibit carcinogenesis. Further precision is

needed to avoid the impression of bias or post-hoc selection of the TS list. How were the tumor suppressors curated? Details of the search and selection strategy are needed.

How do the authors explain the apparent positive and negative correlation between TS and immune inhibitors (line 150 and 153)? Does this imply that some tumor suppressors promote immune evasion? If so, is it accurate to call them tumor suppressors?

The authors should show or describe the individual correlations for the TS they chose e.g. TGFBR2, TSC1 and NF1 from Fig S1F and the immune inhibitors of interest.

Please comment on the choice of 4T1 tumor cell model. Does its known genetic background of mutations <https://www.frontiersin.org/articles/10.3389/fonc.2020.01195/full> potentially affect the impact of TS loss? This is important to generalize the results to other models and patients

Can the authors exclude the possibility that Cas9 expression alone in an immune competent mouse background creates a pro-tumorigenic environment? Either citation or experimental evidence (e.g. Cas9 expression in a nude mouse) would suffice.

Please provide more detail on the India Ink based metastasis nodule counting method. Was this done by a single observer or more? How was a single tumor defined when they appear to grossly abut each other (Fig 3E). What was the reason for not using a histologic measure e.g. tumor cells / lung area? This is important because in Fig 3B the differences in mets per gram appear to be driven largely by outliers and the variance in counts is large. Could this an effect of observer bias?

What was the effect of anti-LAG3 treatment alone? By combining PDL1/LAG3 it is difficult to know whether the reduction in metastases is related to the combination or specific to anti-LAG3 treatment.

In Fig 7E the comparison being made shows that triplet (paclitaxel, LAG3/PDL1) reduces mets per gram but is this specific to TS loss? That is, do TS loss predict sensitivity to paclitaxel + LAG3/PDL1? Could you show this is true by graphing the delta mean met / gram between 4T1-C1 and the TS loss groups for the triplet?

December 8th, 2023

Dear Reviewers,

We greatly appreciate the feedback from all the reviewers. We have conducted additional experiments and analysis to address the reviewers' questions and suggestions. We think this revised manuscript has greatly improved as a result.

A point-by-point response to the reviewers' comments and a list of the incorporated changes are detailed below:

Reviewer #1 (Remarks to the Author):

The authors of this study investigated the role of tumor suppressor inactivation in the modulation of the tumor microenvironment and its impact on immune suppression mediated by LAG3+ T cells. By performing in vivo loss-of-function screening of 102 tumor suppressors in the highly metastatic 4T1 triple-negative breast cancer mouse model, the authors uncovered a novel mechanism in which inactivation of tumor suppressors promotes a non-cell-autonomous inflammatory tumor microenvironment.

This inflammatory microenvironment was found to enhance LAG3+ T cell-mediated immune suppression, providing valuable mechanistic insight into tumor progression and immune evasion. Additionally, the study offers a rationale for patient stratification and potential therapeutic strategies targeting LAG3 in immune checkpoint blockade therapy.

The concept of an inflammatory tumor microenvironment enhancing immune suppression, including LAG3+ T cell-mediated immune suppression, is not entirely novel. Previous studies have reported the involvement of an inflammatory tumor microenvironment in immune suppression and tumor progression. Furthermore, a growing body of literature demonstrating that tumor progression resulting from the downregulation of tumor suppressor genes can be immune-mediated, in addition to being driven by oncogene expression and increased proliferative activity. However, this work would make an excellent addition to this literature and the novelty of the study lies in the specific context of the inactivation of tumor suppressors (i.e. TNBC), the specific findings connecting inactivation of tumor suppressor genes to LAG3+ T cell-mediated immune suppression represent a novel contribution to the field.

The finding that immune therapy targeting LAG3 and PD-L1 decreased metastatic progression of tumors with NF1, TSC1, and TBR1 deficiency is also notable. By showing that targeting both LAG3 and PD-L1 can reduce metastatic progression in this specific context, the study provides valuable information for tailoring immune therapies and potentially improving patient outcomes.

The overall design and approach employed is principled and the methods, including for the

generation and analysis of omics profiling data are suitable. Overall, this is excellent work and is well presented (except, surprisingly, for obvious typos – see below). Also, my comments are relatively minor.

We appreciate the positive feedback and recognition of our novel findings in connecting inactivation of tumor suppressor genes to LAG3+ T cell-mediated immune suppression. We agree with the reviewer that immune therapy targeting LAG3 and PD-L1 should offer additional treatment options for patients whose tumors have NF1, TSC1, and T β RII deficiency, which represent a novel contribution to the field.

Main Comments:

*- It would be appropriate and useful to bring more context in the discussion (see my introductory comments). Prior studies have demonstrated associations between the overexpression of oncogenes and the development of an immunosuppressive microenvironment and should be cited/discussed. e.g. Casey et al. [1], where overexpression of MYC caused by the repression of TS genes was found to contribute to the development of an immunosuppressive TME by upregulating the expression of ICMs CD47 and PD-L1. Or Spranger et al [2], where activation of the β -catenin signaling pathway, which can be triggered by the inactivation of tumor suppressor genes like APC, can lead to an immunosuppressive TME in melanoma. [1] (Casey, S. C., Tong, L., Li, Y., Do, R., Walz, S., Fitzgerald, K. N., Gouw, A. M., Baylot, V., Gütgemann, I., Eilers, M., & Felsher, D. W. (2016). MYC regulates the antitumor immune response through CD47 and PD-L1. *Science*, 352(6282), 227-231) [2] Spranger, S., Bao, R., & Gajewski, T. F. (2015). Melanoma-intrinsic β -catenin signaling prevents anti-tumour immunity. *Nature*, 523(7559), 231-235.*

We appreciate the reviewer's suggestions. The following content and relevant literature have been added in the discussion section to bring more context. "Notably, non-cell-autonomous function of oncogenes have been previously observed ¹. For example, MYC directly regulates CD47 and PD-L1 mRNA expression and suppresses innate and adaptive immune responses in T-cell acute lymphoblastic leukemia ². Melanoma cell-intrinsic activation of β -catenin inhibits CCL4 expression and suppresses recruitment of dendritic cells and T cells, conferring immunotherapy resistance ³. In breast cancer, JAG1-Notch signaling induces the expression of IL-1 β and CCL2, facilitating the recruitment of tumor-associated macrophages in tumors ⁴. These findings, along with our current study, highlight the importance of the non-cell-autonomous functions of TS loss and oncogene activation, as they have a direct impact on the efficacy of immunotherapies by promoting the formation of an immunosuppressive tumor microenvironment."

- The authors do not discuss limitations of their study. This oversight should be rectified.

In response to the reviewer's criticism, we have added a discussion of several limitations: "These findings have yet to be verified in other cancers or different breast cancer subtypes. TSs vary among cancer types, with each exhibiting distinct TS loss and oncogene mutations/overexpression. Further, in addition to shared downstream molecules resulting

from the loss of NF1, TSC1, or TβRII, we also identified factors unique to specific TS loss, which will be of future research interest. Lastly, dissecting the molecular mediators underlying TS regulation of the chromatin landscape should offer more mechanistic insight.”

- RNA-seq and ATAC-seq data have not been deposited in public repositories. It should be the case for transparency-sake and to align with open data standards. Failing to do so would go against the FAIR principles for data management and sharing and significantly reduce the long-term impact and interest in the study.

All the RNA-seq and ATAC-seq data sets will be deposited at the acceptance of the manuscript.

- While the authors have taken care to minimize off-target effects during library design (ranking sgRNAs by an off-target score using a metric), they do not mention including non-targeting (scrambled) sgRNA controls to account for potential off-target effects.

We apologize for this omission. Our sgRNA library includes 100 non-targeting (scrambled) sgRNA controls designed to have minimal homology to sequences in the mouse genome, and 40 positive control sgRNAs targeting 40 genes in which loss of function of these genes is lethal to cells. This information was added into the methods section, and their sequence information was uploaded together with the experimental genes.

Minor comments

- It is not clear why the color scheme for TS-high vs TS-low patients changes between panels C and D on figure 1 (or E and F in Figure S1).

The different color scheme for each figure is intended to show results from different human datasets, for results consistency.

- Font size on the volcano plot Figure 5S is impossibly small

The font size is increased.

*- There are some typos / incomplete sentences here or there. Check for instance:
o Line 164: sentence construction, missing word?*

Correction is made. Thanks.

o Line 165 and other places: neutrophiles

Changed to neutrophils.

o Line 181: sentence construction.

Correction is made.

o Fig S5 legend: "balk" is presumably meant as "bulk"

Correction has been made. Thanks

- Just a comment, but it makes it much easier to review a manuscript when the legends are embedded with the figures (as you have done for supplementary figures).

Thanks for the suggestions.

Reviewer #2 (Remarks to the Author):

In the manuscript, "Loss of tumor suppressors promotes a non-cell-autonomous inflammatory tumor microenvironment and enhances LAG3+T cell mediated immune suppression" Zahraeifard and colleagues aim to understand how loss of expression of tumor suppressors contributes to an altered immune microenvironment and influences response to anti-cancer immunotherapy.

They use a combination of human datasets with cell line xenograft and syngeneic mouse tumor models to perform a CRISPR screen and identify loss of which tumor suppressors is associated with increased number of lung metastases. They then use RNA seq and ATAC seq to show that the genes altered by TS loss are in the STAT pathway. Using flow cytometry, CODEX and CYTEK on transgenic mice they find that TS deficient tumors are associated with increased LY6G+ neutrophils and LAG3+ T-cells. They then combine paclitaxel, anti Lag3 and PDL1 targeting therapy to show that response to this combination is increased in TS lost tumors compared to control.

The authors have tackled an important question: how does the loss of tumor suppressors affect the anti-tumor response. As this is a complicated multi-step process they have shown evidence from each layer of the 'system' to support their argument. However, their claim that "a direct and causal" [line 395] inflammatory or tissue damage response in response to TS loss is not supported by the data. Instead, what they show is several associations without clear mechanistic connection between each phenotype e.g. TSC1 loss and increased tumor growth and associated STAT pathway activation and associated LAG3+ T-cell infiltrate. They have not shown how TSC1 loss is necessary and sufficient to alter the chromatin landscape within tumor cells resulting in JAK-STAT pathway activation and consequent alteration in the tumor microenvironment. In fact, given that loss of TSC1, TGFBR2 and NF1 lead to common ATAC changes in IL6 and JAK3, is it instead more likely that a separate downstream activator is causally responsible for this change in expression program? One way to demonstrate that this is a specific effect causally related to tumor suppressor loss is to show that phenotype can be rescued. For example, for tumors with TSC1 loss, does mTOR inhibition blunt tumor growth by preventing differential expression of JAK-STAT pathway genes? Can they further show this is a

specific effect to the loss of their selected tumor suppressors (e.g. NF1) by using their elegant negative control sgNF2? For example, if sgNF1 tumors are compared to sgNF2, does the JAK-STAT pathway still appear to be responsible for the difference in tumor metastasis burden? Is there also a difference in LAG3+ T-cell frequency? Do sgNF2 tumors respond less well than sgNF1 tumors to treatment with paclitaxel and PDL1/LAG3? This is an important question to determine the specificity of the TS loss phenotype.

We thank the reviewer for the positive feedback that our manuscript tackles an important question. We appreciate the reviewer's suggestions, which we found helpful. We thus performed an animal experiment to investigate the causal role of JAK-STAT pathway activation and increased LAG3+ T cell downstream of TS loss by comparing 4T1-C1, 4T1-sgNf1 or 4T1-sgNf2 cells.

First, we examined mTOR, a direct downstream target of TSC1. As shown in Western below, STAT6, but not mTOR activation was increased comparing 4T1-sgTsc1 with 4T1-C1 control cells. In addition, several studies have reported mTOR-independent functions of TSC1^{5,6}. Thus, the activation of JAK/STAT signaling is likely a mTOR signaling-independent function downstream of TSC1 loss in our context.

Second, as suggested by the reviewer, we used Tofacitinib in a rescue experiment to investigate the specific effect (JAK activation and increased LAG3+ T cells) causally related to tumor suppressor loss. Tofacitinib is an FDA-approved pan-JAK inhibitor that preferentially targets JAK3 and JAK1; it is extensively used in preclinical studies, including those focused on cancer⁷⁻¹⁰. Our results show that Tofacitinib treatment significantly reduced metastasis in 4T1-sgNf1 cells, but caused no significant change in the metastasis of 4T1-C1 and 4T1-sgNf2 cells. In addition, Tofacitinib reduced LAG3+ T cells only in 4T1-sgNf1 tumors, but not in 4T1-sgNf2 tumors. The new data is included in Figure 6I and J, and the manuscript text and figure legend have been updated accordingly.

Third, we performed an additional immunotherapy experiment using Nf2 knockout cells as a control. The combined treatment of paclitaxel with anti-LAG3 and anti-PDL1 significantly decreased metastasis from tumors with NF1 loss, but not with NF2 loss. The manuscript has been updated with this new result in Fig S6C.

Figure legend: increased pSTAT6 was found in 4T1-sgNf1, 4T1-sgTsc1, and 4T1-sgTgfr2 cells in comparison to 4T1-C1 cells, while no significant changes in p-mTOR or pS6K1.

The manuscript would be further improved by making the following minor revisions:

The authors have curated a list of 102 proteins and defined them as tumor suppressors. Each has a different function and many works in concert to inhibit carcinogenesis. Further precision is needed to avoid the impression of bias or post-hoc selection of the TS list. How were the tumor suppressors curated? Details of the search and selection strategy are needed.

- (1) We use mouse models of breast cancer metastasis, and our readout is metastatic nodule counts. The 4T1 cancer cells are aggressive and very metastatic. It is counterproductive to perform a genome-wide CRISPR in vivo screens, which would require an injection of 1×10^7 cells¹¹. With injection of 1×10^7 cells, the mice would not survive long enough for proper evaluation of metastatic colonization. Our 4T1 model of metastasis utilizes 2×10^5 cells for mammary fat pad injection. The tumor progression and lung metastasis in this model are well characterized and widely used in the field.**
- (2) Genome-wide human and mouse CRISPR libraries typically include >3–4 sgRNAs per gene with sgRNA representation at 500x¹². In the current study, the sgRNA library was designed as previously described and included 6 sgRNAs per gene at 270x representation¹³. As shown in Fig. 3, 3-4 sgRNAs were identified per gene (TgfbR2/TSC1/NF1/NF2) in lung nodules. The identification of multiple sgRNAs per gene increases our confidence in their roles in metastasis.**
- (3) Based on these considerations, we curated an sgRNA list of ~100 genes. It covers three major categories: (1) Metastasis suppressors that were identified in cancers in addition to breast cancer; (2) Epigenetic modifiers whose functional loss promotes cancer progression and metastasis; (3) Classic tumor suppressors known in cancer biology field literature. To ensure their tumor suppressor function, these genes were experimentally validated and confirmed by more than two references.**

How do the authors explain the apparent positive and negative correlation between TS and immune inhibitors (line 150 and 153)? Does this imply that some tumor suppressors promote immune evasion? If so, is it accurate to call them tumor suppressors?

This is indeed an interesting observation. Tumor suppressors have different mechanisms of function, where some acting mostly as cells-autonomous signaling regulators, while others play critical roles in inflammatory and immune homeostasis. In addition, the context of other TSs and even oncogenes may also play roles. Clearly, further data analysis and experimental validation will be needed to get a clear answer.

In the current study, our data reveal loss of several TSs exhibiting immune evasion phenotype and are supported by additional data analysis and experimental evidence.

The authors should show or describe the individual correlations for the TS they chose e.g. TGFB2, TSC1 and NF1 from Fig S1F and the immune inhibitors of interest.

Yes, per the reviewer request, we used the Nature Medicine human scRNA-seq dataset and performed additional analyses to investigate the correlation between the individual TS (TSC1, NF1 and TGFBR2) and T cell immune inhibitory markers. TSC1 and NF1 showed a significant inverse correlation with PD1⁺/LAG3⁺ T cells, whereas TGFBR2 followed a similar but non-significant trend. The results are presented below for the reviewer. We want to caution that this type of analysis may be limited due to a short list of database and patient number. This result is consistent with the inverse correlation of individual TSs and LAG3 from the TCGA dataset analysis presented in left panel of Fig. 7D.

A Nature Medicine, 2021

Please comment on the choice of 4T1 tumor cell model. Does its known genetic background of mutations <https://www.frontiersin.org/articles/10.3389/fonc.2020.01195/full> potentially affect the impact of TS loss? This is important to generalize the results to other models and patients.

4T1 is one of the most extensively used mouse models for spontaneous metastasis of TNBC. Our lab and others have also used it for immunotherapies, especially in studying immune inhibitors. The most notable genetic mutation in the 4T1 cell line is the TP53 null mutation, which is appropriate given that TP53 mutations are highly prevalent (60%) in TNBC patients. The 4T1-derived C1 clonal cell line was used as a control throughout the TS KO experiments. The results are specific to TS loss and does not seem to suggest that p53 has an impact on the consequences of TS loss.

Can the authors exclude the possibility that Cas9 expression alone in an immune competent mouse background creates a pro-tumorigenic environment? Either citation or experimental evidence (e.g. Cas9 expression in a nude mouse) would suffice.

We use Cas9 transgenic mice that constitutively express Cas9, thus minimizing immunogenicity against the Cas9 protein¹⁴. This mouse line is widely used for *in vivo* studies involving CRISPR-Cas9 gene editing¹⁵⁻¹⁸.

In the study by Ajina et al., immune profiling of spleen from Cas9 transgenic and wild-type mice showed no significant difference in the percentages of T cells, B cells, NK cells, or

myeloid cells¹⁷. Chu et al., using a different line of Cas9 transgenic line, also found no altered immune profiling, including HSCs, B cells, T cells, and myeloid cells¹⁹.

Please provide more detail on the India Ink based metastasis nodule counting method. Was this done by a single observer or more? How was a single tumor defined when they appear to grossly about each other (Fig 3E). What was the reason for not using a histologic measure e.g. tumor cells / lung area? This is important because in Fig 3B the differences in mets per gram appear to be driven largely by outliers and the variance in counts is large. Could this an effect of observer bias?

The counts were obtained by a single observer, and sometimes were verified by a second person when the counts were not as clear or easy. In cases where nodules were in close contact, other factors such as their shape, edge, and the size, could be taken into consideration. Most importantly, the quantitative analysis was performed in a similar manner throughout the study. Indian ink is a well-established procedure in the field. We have used it to count lung metastatic nodules in several publications (e.g., Park WY, et al. and Yang L. *Nat Cancer*. 2023 PMID: 36973439; PMCID: PMC10042736; So JY, et al. and Yang L. *Cancer Res*. 2020 PMID: 32265226; PMCID: PMC7299749).

A histologic measure often gives the nodule counts in a single section, which may not be the best representation of the whole lungs. In addition, the large number of mice used for each set of experiments (at least 30 mice per experiment) makes Indian ink more practical in terms of cost and time.

We are aware of the large variations in metastatic nodule counts shown in Fig 3B. However, this is a common observation among many laboratories that study spontaneous metastasis. The contributing factors include individual animal variability, tumor cell heterogeneity, and multiple metastatic steps, etc. Please be reminded that the results were verified in an independent experiment using shRNA (Figure 3E).

What was the effect of anti-LAG3 treatment alone? By combining PDL1/LAG3 it is difficult to know whether the reduction in metastases is related to the combination or specific to anti-LAG3 treatment.

To answer the reviewer's question, we conducted an additional animal experiment including a treatment group with the combination of paclitaxel and anti-LAG3 treatment. The paclitaxel + anti-LAG3 treatment reduced metastasis in 4T1-sgTSC1 and 4T1-sg NF1 tumors. However, the combination treatment of paclitaxel with anti-LAG3/PDL1 showed the most significant reduction of metastasis. As the groups treated with paclitaxel alone and the combination of paclitaxel with anti-LAG3/PDL1 treatment showed consistent results with the previous results, they were added to the updated Fig. 7E and Fig. S6B.

In Fig 7E the comparison being made shows that triplet (paclitaxel, LAG3/PDL1) reduces mets per gram but is this specific to TS loss? That is, do TS loss predict sensitivity to paclitaxel + LAG3/PDL1? Could you show this is true by graphing the delta mean met / gram between 4T1-C1 and the TS loss groups for the triplet?

We appreciate the reviewer's suggestion. To analyze the specificity of TS loss in the reduction of metastasis by the combination treatment of paclitaxel with anti-LAG3/PDL1, each met count/gram tumor value was converted to a relative value against the average met count/gram tumor in the control treatment group of each TS loss. The result showed significant difference in the relative reduction of met count/gram tumor by the combination treatment in all three TS loss groups when compared to 4T1-C1 group as shown below.

References

- Wellenstein, M.D., *et al.* Loss of p53 triggers WNT-dependent systemic inflammation to drive breast cancer metastasis. *Nature* **572**, 538-542 (2019).
- Casey, S.C., *et al.* MYC regulates the antitumor immune response through CD47 and PD-L1. *Science* **352**, 227-231 (2016).
- Spranger, S., Bao, R. & Gajewski, T.F. Melanoma-intrinsic beta-catenin signalling prevents anti-tumour immunity. *Nature* **523**, 231-235 (2015).
- Shen, Q., *et al.* Notch Shapes the Innate Immunophenotype in Breast Cancer. *Cancer Discov* **7**, 1320-1335 (2017).
- Alves, M.M., *et al.* PAK2 is an effector of TSC1/2 signaling independent of mTOR and a potential therapeutic target for Tuberous Sclerosis Complex. *Sci Rep* **5**, 14534 (2015).
- Lai, M., *et al.* Tsc1 regulates tight junction independent of mTORC1. *Proc Natl Acad Sci U S A* **118**(2021).
- Jiang, J.K., *et al.* Examining the chirality, conformation and selective kinase inhibition of 3-((3R,4R)-4-methyl-3-(methyl(7H-pyrrolo[2,3-d]pyrimidin-4-yl)amino)piperidin-1-yl)-3-oxopropanenitrile (CP-690,550). *J Med Chem* **51**, 8012-8018 (2008).
- Sandborn, W.J., *et al.* Tofacitinib, an oral Janus kinase inhibitor, in active ulcerative colitis. *N Engl J Med* **367**, 616-624 (2012).
- Simon, N., Antignani, A., Hewitt, S.M., Gadina, M., Alewine, C. & FitzGerald, D. Tofacitinib enhances delivery of antibody-based therapeutics to tumor cells through modulation of inflammatory cells. *JCI Insight* **4**(2019).
- Tanaka, Y., Luo, Y., O'Shea, J.J. & Nakayamada, S. Janus kinase-targeting therapies in rheumatology: a mechanisms-based approach. *Nat Rev Rheumatol* **18**, 133-145 (2022).
- Chen, S., *et al.* Genome-wide CRISPR screen in a mouse model of tumor growth and metastasis. *Cell* **160**, 1246-1260 (2015).

12. Doench, J.G. Am I ready for CRISPR? A user's guide to genetic screens. *Nat Rev Genet* **19**, 67-80 (2018).
13. Read, A., Gao, S., Batchelor, E. & Luo, J. Flexible CRISPR library construction using parallel oligonucleotide retrieval. *Nucleic Acids Res* **45**, e101 (2017).
14. Platt, R.J., *et al.* CRISPR-Cas9 knockin mice for genome editing and cancer modeling. *Cell* **159**, 440-455 (2014).
15. Carroll, K.J., *et al.* A mouse model for adult cardiac-specific gene deletion with CRISPR/Cas9. *Proc Natl Acad Sci U S A* **113**, 338-343 (2016).
16. Zhao, X.Y., *et al.* Long noncoding RNA licensing of obesity-linked hepatic lipogenesis and NAFLD pathogenesis. *Nat Commun* **9**, 2986 (2018).
17. Ajina, R., *et al.* SpCas9-expression by tumor cells can cause T cell-dependent tumor rejection in immunocompetent mice. *Oncoimmunology* **8**, e1577127 (2019).
18. Liu, Q., *et al.* Cerebellum-enriched protein INPP5A contributes to selective neuropathology in mouse model of spinocerebellar ataxias type 17. *Nat Commun* **11**, 1101 (2020).
19. Chu, V.T., *et al.* Efficient CRISPR-mediated mutagenesis in primary immune cells using CrispRGold and a C57BL/6 Cas9 transgenic mouse line. *Proc Natl Acad Sci U S A* **113**, 12514-12519 (2016).

REVIEWERS' COMMENTS

Reviewer #1 (Remarks to the Author):

After careful consideration of the author's response to the reviewers' comments and the revisions made to the manuscript titled "Loss of tumor suppressors promotes a non-cell-autonomous inflammatory tumor microenvironment and enhances LAG3+T cell mediated immune suppression," I am pleased to report that the author has demonstrated a commendable effort in addressing the feedback provided by the reviewers. The revised manuscript reflects a significant improvement in both content and presentation.

The inclusion of relevant literature in the discussion provides a broader context and highlights the novelty of the findings, particularly in the area of tumor suppressor inactivation and its impact on LAG3+ T cell-mediated immune suppression in triple-negative breast cancer.

Furthermore, the author's acknowledgment of the study's limitations and their commitment to depositing RNA-seq and ATAC-seq data sets in public repositories upon acceptance of the manuscript align with best practices for transparency and data sharing. The attention to detail in addressing specific concerns, such as the inclusion of control experiments and the clarification of figures and text, enhances the manuscript's clarity and readability.

Overall, the revised manuscript represents a valuable addition to the field, and the author's diligent work in refining their study is commendable. I believe that the manuscript, in its current form, is well-prepared for publication and will make a significant contribution to the scientific community.

Reviewer #2 (Remarks to the Author):

The authors have thoroughly and thoughtfully addressed the revision points. The manuscript is stronger for it. Well done.